# communications
# earth & environment

# Plastic pollution in riverbeds fundamentally affects natural sand transport processes

Catherine E. Russell ⓘ [1,2,3✉], Roberto Fernández ⓘ [4,5], Daniel R. Parsons ⓘ [5,6] & Sarah E. Gabbott ⓘ [1]

Over the past 50 years, rivers have become increasingly important vectors for plastic pollution. Lowland riverbeds exhibit coherent morphological features including ripple and dune bedforms, which transport sediment downstream via well-understood processes, yet the impact of plastic on sediment transport mechanics is largely unknown. Here we use flume tank experiments to show that when plastic particles are introduced to sandy riverbeds, even at relatively low concentrations, novel bedform morphologies and altered processes emerge, including irregular bedform stoss erosion and dune "washout", causing topographic bedform amplitudes to decline. We detail (i) new mechanisms of plastic incorporation and transport in riverbed dunes, and (ii) how sedimentary processes are fundamentally influenced. Our laboratory flume tank experiments suggest that plastic is not a passive component of river systems but directly affects bed topography and locally increases the proportion of sand suspended in the water column, which at larger scales, has the potential to impact river ecosystems and wider landscapes. The resulting plastic distribution in the sediment is heterogeneous, highlighting the challenge of representatively sampling plastic concentrations in river sediments. Our insights are part of an ongoing suite of efforts contributing to the establishment of a new branch of process sedimentology: plastic – riverbed sand interactions.

[1] School of Geography, Geology, and the Environment, University of Leicester, Leicester LE1 7RH, UK. [2] Department of Geology and Geophysics, Louisiana State University, Baton Rouge, LA 70803, USA. [3] University of New Orleans, New Orleans 2000 Lakeshore Drive, LA 70148, USA. [4] Department of Civil and Environmental Engineering, Penn State University, State College, University Park, PA 16802, USA. [5] Energy and Environment Institute, University of Hull, Hull HU6 7RX, UK. [6] Loughborough University, Loughborough LE11 3TU, UK. ✉email: catherine.russell@fulbrightmail.org

Plastics, and their derived products, are pervasive within earth surface systems globally; they are in the air that we breathe[1,2], in agricultural soils[3,4], in aquatic biota[5,6], throughout fresh and marine waterbodies[7–9], and in the deepest abyssal trenches[10,11]. Rivers are the primary terrestrial conduits of plastic, carrying and delivering an estimated 0.8–2.7 million metric tonnes of plastics to coastal and marine environments each year[12]. Yet, despite advancements in understanding how plastic particles settle[13,14], are eroded from the riverbed[15], travel in rivers[16–18], and through the porous networks formed within particle deposits[19], our understanding of their impact on broader sediment transport processes and sedimentary systems more generally is in its infancy[20]. Their presence and interactions have implications for environmental monitoring, representative sampling, sediment transport processes, riverbed morphodynamics, landscape evolution, and sedimentary geology.

Despite a high variability in river shapes globally[21,22], dunes are ubiquitous features on their mobile riverbeds[23–25]. River dunes, which have a shallow upstream stoss side, and a steeper downstream lee-side, migrate downstream by the continuous erosion of sediment particles from the stoss-side and subsequent deposition of material on the lee side[26,27] (Fig. 1a). The downstream migration of these riverbed dunes can influence the shape of the wider river channel[28], and they are important in controlling and maintaining sediment flux that is crucial for a suite of ecosystem services[29,30].

Bedform geometries and processes are predominantly dictated by the grain-to-grain interactions between the sediment particles and the flow. However, small-scale localised changes caused by biota can compound to result in large-scale and substantial wide-reaching effects on landscape dynamics[31–33]. Biotic processes alter and interact with bedform dynamics[34–36], altering topographies[33,37,38], grain size distributions[39], and particle cohesion[35,40]. Additionally, organic material such as log jams[41], and the deposition of water-logged charcoal from forest fires may influence river processes and the wider landscape[42].

However, organic materials are part of the natural workings of rivers, and have been for around 400 million years, so whilst these elements are important to understand, they do not represent a change in how rivers are responding to pollution. Whilst both organic material and most plastics have comparatively low density to riverbed sediment, they are not analogous as plastic properties are far more variable. As such, further studies are needed to appropriately constrain the similarities and differences in transport mechanics and riverbed disruption, and studies of organic material may not be directly translated to plastic studies. Additionally, plastic particles are increasingly abundant and will outlive organic debris in the environment for perhaps centuries[43]. Therefore, plastic particles have the potential to accumulate and disrupt sedimentary mechanics over a longer period, leading to wide-reaching and long-lasting changes to fundamental grain-to-grain interactions in sediment transport.

Plastic has a wide range of physical properties, morphologies, and characteristics[44], such that particles less dense than fresh water will travel floating on or near the water surface, whereas denser particles will travel closer to the riverbed, or even bounce or roll along it[45]. A substantial volume of plastic manufactured is denser than freshwater[46], resulting in large quantities of plastic travelling under the surface waters of rivers[45,47] and interacting with, or being temporarily stored in, the riverbed[17,48,49]. Observing and sampling riverbed processes in natural environments without disturbing them is challenging, and thus monitoring and characterising the effect of plastic on riverbed transport processes in the field is difficult. Experimental flume tanks are a successful and long-established approach to establish robust and comparable data for natural systems including rivers, deltas,

and estuaries[50]. Therefore, we designed and undertook physical laboratory experiments in a 10 m long, 0.5 m wide recirculating flume. We added enough sand to create a 0.1 m thick deposit over the entire length of the flume (1,241.6 kg) with a median size ($D_{50}$) of 0.23 mm. The flume was filled with water to a depth of 0.2 m above the sand bed and turned on to convey flow at a mean velocity of 0.5 m/s. After the sand bed had developed dunes that were in equilibrium with the flow (12 h) we added a mixture of plastic materials (1.49 kg) of different densities and sizes (Table 1) and documented the interactions between the plastic particles and the sand particles over the channel bed for >12 h. These experiments were designed to explore whether plastic influences riverbed sand transport mechanisms and focused on representing systems such as small urban stream systems, which are commonly highly contaminated with plastics of different types due to their proximity to pollution sources[12]. A full description of the materials and setup is available in the Methods section.

Through contaminating sand dunes with plastic in a recirculating laboratory flume tank, we test how the inclusion of plastic particles in riverbed sand dunes impacts their shapes (morphologies) and migration processes. Using small-scale laboratory experiments to study geomorphic and sedimentary processes has long proved to be an"unreasonably-effective" method capable of reproducing field observations despite differences in spatio-temporal scales, as the underlying physics are captured due to natural scale independence, also called morphodynamic similarity[50]. Our experiments explore how and where plastics interact with bedform dynamics, and how these interactions impact a host of processes known to be important for sediment transport and hydrology, as well as the resulting sedimentary deposit. We seek to understand what the results mean for the fate and distribution of plastic and demonstrate important implications for representative sampling of plastic in riverbeds, as well as understanding the controls on where plastics may be concentrated in certain environments, thereby aiding understanding for environmental monitoring studies. We advocate that our results offer fundamental insights on what we suggest is a new sub-branch of sedimentology: sediment and anthropogenic particle interactions, set to be increasingly relevant as earth scientists prepare to describe, interpret, and understand Anthropocene landscapes.

## Results and discussion

River dunes typically migrate by progressive erosion of sand grains from the stoss side and deposition on the lee side (Fig. 1a). Overall, flow patterns in river channels are affected by the presence of dunes, with higher flow velocities above the dune crest and slower flow velocities above the dune trough[51] (Fig. 1ai). The dune shape itself often leads to flow separation (detachment) and a recirculation zone within the dune lee side, where flow velocities reduce, and sediment grains tend to be deposited[52]. The results herein show that the well-established flow and sediment transport processes are impacted by the presence of plastic particles, which often have lower density than sand and therefore are easier to entrain and be transported by the flow.

Sand (median sediment size $D_{50}$ of 0.23 mm) and 13 different types of plastic were put into a recirculating laboratory flume tank under a constant flow discharge (of 0.05 m³/s) to observe interactions between the plastic and sand under transport conditions (see "Methods" for full details and Supplementary Movie 1 for footage of the experiment). The experimental conditions were selected as they produce appropriately scaled dunes for the pollutants, i.e., the dune height ought to exceed the height of the largest particle, such that it may be incorporated in the

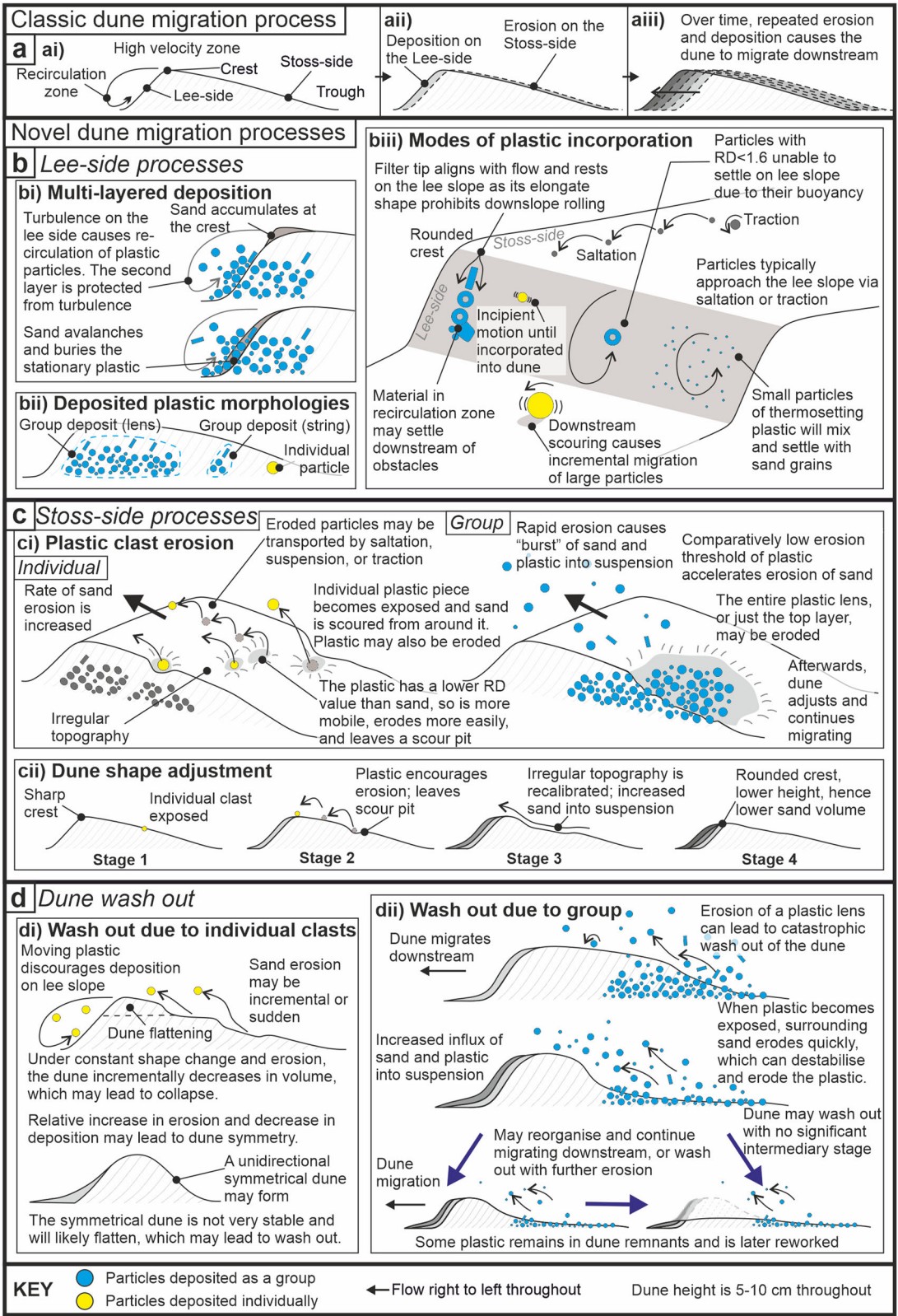

**Fig. 1 Processes by which sand is included in, and subsequently eroded from, small sand dunes. a** The standard terminologies and classic dune migration process; **b** Lee-side processes in a plastic polluted sand dune; **c** Stoss-side processes in a plastic polluted sand dune and consequential eroded profiles; **d** Processes of dune wash-out whereby the extent of disturbance of the dune by the plastic is too great for the dune to sustain its downstream migration.

bedform, and the flow speed ought to be rapid enough to mobilize the particles. As this is an emerging discipline in understanding how plastic and sand interact on a riverbed, all insights of plastic and sediment interactions are novel and of value. The size (i.e.,

equivalent diameter given in mm), (D) and submerged density (i.e., density of the object minus the density of water g/cm³) (R) of the plastic and the sand defines their mobility, thereby their erosion and entrainment thresholds. All plastics used herein had

**Table 1 A table of included plastic particles and their properties.**

| Particle | Material | Mass[c] | Density | Equivalent sphere | Submerged specific | Product | RD ratio to | Observed |
| | | (g) | ρ (g/cm³) | Diameter D (mm)[a] | Gravity R (-) | RD (mm) | sand (-) | Mobility |
| --- | --- | --- | --- | --- | --- | --- | --- | --- |
| 4 mm bead | Polystyrene | 38.2 | 1.05 | 4.0 | 0.05 | 0.20 | 1.90 | High |
| 8 mm pony bead[d] | Polystyrene | 127.6 | 1.05 | 8.0 | 0.05 | 0.40 | 0.95 | High |
| 6 mm bb gun pellet | Acrylonitrile butadiene styrene | 199.8 | 1.05 | 6.0 | 0.05 | 0.30 | 1.27 | High |
| 7 mm Mardi Gras bead[d] | Polycarbonate | 132.2 | 1.20 | 7.0 | 0.20 | 1.40 | 0.27 | Low |
| 9.5 mm Mardi Gras bead[d] | Polycarbonate | 50.8 | 1.20 | 10 | 0.20 | 1.90 | 0.20 | Low |
| 12 mm Mardi Gras bead[d] | Polycarbonate | 198.2 | 1.20 | 12 | 0.20 | 2.40 | 0.16 | Low |
| 14 mm Mardi Gras bead[d] | Polycarbonate | 85.7 | 1.20 | 14 | 0.20 | 2.80 | 0.14 | Low |
| 1 × 1 cm polyester fabric | Polyester | 8.3 | 1.38 | 2.7 | 0.38 | 1.02 | 0.37 | High |
| 1 × 1 cm fleece | Polyethylene terephthalate | 22.5 | 1.38 | 2.7 | 0.38 | 1.02 | 0.37 | High |
| 1 × 1 cm baby wipe | Polypropolene | 15.5 | 1.92 | 2.7 | 0.92 | 2.46 | 0.15 | Low[b] |
| 20 × 8 mm cigarette filter tip[d] | Cellulose acetate fibre | 8.8 | 1.33 | 10 | 0.33 | 3.31 | 0.11 | Low |
| 0.3 mm PVC powder | PVC | 200.0 | 1.36 | 0.30 | 0.36 | 0.11 | 3.51 | High |
| 1 mm plastic fragments[d] | Recycled, ground melamine | 399.6 | 1.6 | 1.0 | 0.6 | 0.60 | 0.63 | High |
| 0.23 mm sediment | Sand (quartz) | 1241600 | 2.65 | 0.23 | 1.65 | 0.38 | 1.00 | High |

[a]Diameter for non-spherical particles obtained by computing the particle volume and solving for the diameter of the equivalent sphere.
[b]Behaviour of baby wipes varied due to their properties; they did not maintain their original shape, were torn apart and absorbed fine sediment grains.
[c]Mass refers to total mass of all particles of a plastic particle type.
[d]Interacts with bed material.

a lower density than sand (Table 1), however, some had larger sizes and therefore comparatively lower mobility, leading to a range of complex interactions. The results show that, for the flow conditions in the flume tank and the particles used, their behaviour and interaction with the sand bed changed at RD ~ 1 mm, (where RD is the particle submerged specific gravity –R- multiplied by the particle diameter –D). Particles with RD > 1 were found to be less mobile and more readily incorporated into the sand dune, than particles with RD < 1, which had greater mobility and were less readily incorporated into the sand substrate (Table 1).

**Lee-side processes**. Migrating sand dunes were found to trap plastics that accumulated on the lee side of the dunes and these particles were incorporated into the substrate, sometimes as a group (or lens) of particles, and sometime as individual particles (Fig. 1b). Plastic particles reached the lee slope via saltation, or traction transport up the stoss side of the dune. Upon reaching the crest, plastic particles would readily re-enter suspension in the high-velocity zone, roll down the lee slope, or enter the recirculation zone on the lee front. Sand particles accumulate at the crest and periodically avalanche down towards the lee trough, where the flow is slower and less turbulent. Plastic with high specific gravity, and thus lower mobility (see Methods and Table 1), was found to accumulate on the lee side of the dune, where it exhibited intermittent incipient motion, influenced by well-known flow pulses found within the recirculation zone. These particles remained in-situ until avalanching sand grains buried the plastic as the dune migrated downstream. Plastic was deposited as individual particles, or as components of a group deposit that may itself be a lens or a string (see Fig. 1bii). Plastic with low specific gravity (i.e. <1 RD), and thus higher mobility (see Methods and Table 1), is easier to entrain in the flow, and thus require sheltering from the turbulence within the recirculation zone to become incorporated into the dune substrate.

Deposition of plastic with low specific gravity happens either by: (i) multi-layering due to high local volume of plastic accumulated on the lee-side, which can form a lens if the multi-layered deposition process is prolonged, and an isolated layer, or string, if short-lived (Fig. 1bii); or (ii) due to an obstacle higher on the slope (e.g. cigarette filter tips), which enables down-slope deposition as part of a lens or as an isolated string (Fig. 1biii). Therefore, multi-layered deposition (see Figs 1bi; 2a) occurs when there is a large enough accumulation to form two or more layers of plastic particles on the lee slope. The uppermost layer interacts with the flow in the recirculation zone, sheltering the bottom layer for long enough to increase the chance that these particles are incorporated into the dune substrate by periodic lee-side sand avalanches.

Cigarette filter tips aligned with the flow such that they settled and did not roll down the lee slope, hence forming obstacles that presented zones of sheltered deposition down slope. As such, filter tips were commonly accompanied by a tail of plastic particles (Fig. 1biii), highlighting the importance of plastic shape on their behaviour and the characteristics of the depositional fate. The largest spherical plastic particles in the experiment (14 mm Mardi Gras beads) rolled down the lee slope largely unaffected by the recirculation zone turbulence. The size of the 14 mm Mardi Gras beads was sufficient to cause local flow acceleration zones that promoted scour downstream of the obstacle it formed, hence the particle could roll further downstream, before eventually being incorporated at the base of the lee slope at the point of lowest scour depth.

Plastics with RD > 1 were more readily incorporated in the lee slope, as they were able to hold their position between occurrences of sand avalanche, which subsequently secured the plastic particle in the deposit. Finally, plastic with similar mobility to the sand grains, such as the 1 mm plastic fragments (Table 1), were found to settle out of the recirculation zone steadily, punctuated by sand avalanches, generating a distinctive diagonally layered deposit characteristic seen throughout the deposits (Fig. 3).

Our results suggest that plastics can interact with the bed, and deposit in the lee-side of dunes even under steady flow, whereas

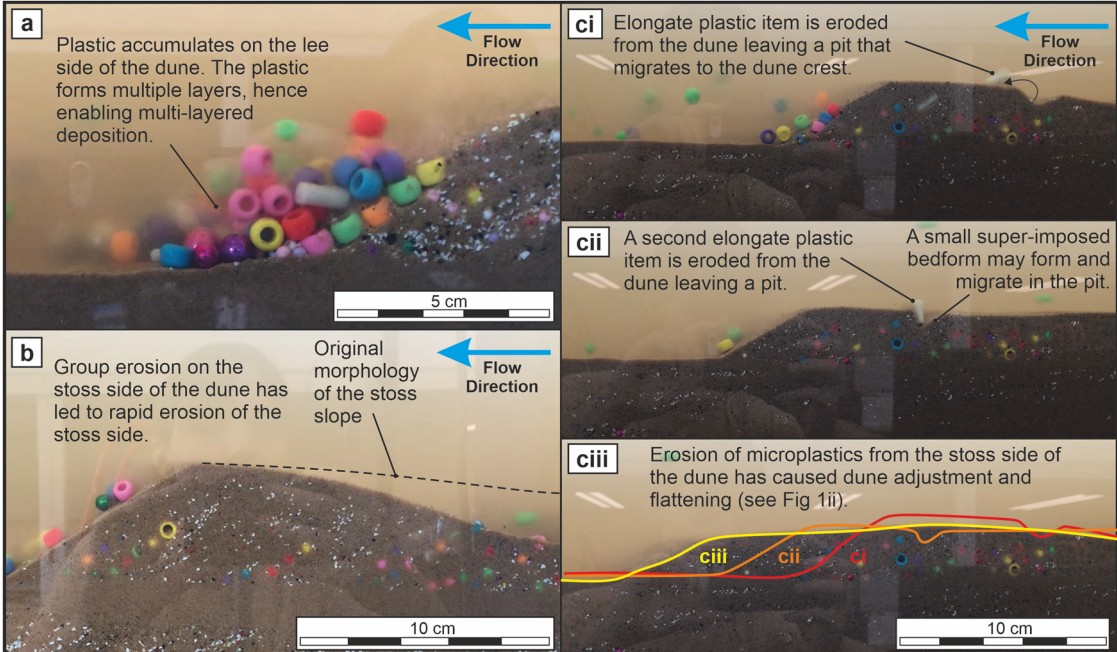

**Fig. 2 Images of the plastic particle interactions in sand dunes during the experiment. a** Multi-layered deposition; **b** A dune that has undertaken group erosion on the stoss-side and is now equilibrating; **c** The erosion of filter tips in ci and cii leads to distinctly changed dune morphology.

many reports on organic materials require changes in flow velocities, which are common during tidal cycles in estuaries[53,54]. However, the deposition of organic debris such as water-logged charcoal[42], has been documented to occur in the troughs of bedforms in constant flow, giving rise to morphologies similar to the "lenses" we describe as resulting from plastic transport (see Fig. 1bii).

**Stoss side processes**. As the dunes migrated downstream, the incorporated plastic became exposed on the stoss side and was eroded rapidly due to the lower RD value, the roughness difference, and the size difference between the plastic and the sand. The lower RD value of plastic compared to sand corresponds to its relatively lower erosion threshold, and thus its relative higher mobility (Fig. 1c, Table 1). The roughness difference, and the size difference between the plastic and the sand encourages erosion of the sand around the plastic obstacle due to local flow acceleration. However, it is not necessary for the plastic to be removed from the stoss side of the dune for the dune morphology to be affected, as a scoured pit may form and persist due to the particle properties difference alone.

Yet, when individual plastic pieces are eroded during migration of the bedform, they may be re-entrained via traction, saltation, or suspension, and leave a pit on the stoss side of the dune (Fig. 1ci). The erosion and rapid re-entrainment of plastic created small depressions, pits, or scours in the stoss side of the sand dune that subsequently migrated to the dune crest and sometimes induced the generation of small super-imposed bedforms along the stoss[23,55] (Fig. 2cii). The pits were observed to become larger and migrate towards the dune crest resulting in dunes with a more rounded crest, a lower overall height, and lower sand volumes, and thus general flattening of the topography, i.e., diminished dunes[56] (Figs. 1cii; 2ci-iii). In cases where a plastic lens is eroded, the plastic may be eroded rapidly from the sand in a "burst" of sediment and plastic that is eroded from the dune and re-entrained in the flow (Fig. 1ci). Such bursts of eroded material were found to lead to a steeper stoss side, which encouraged further erosion of sand and plastic as the dune sought to

re-establish equilibrium (Figs 1ci; 1dii; 2b). The upper portion of the plastic lens, or indeed the entire lens, may be eroded and the affected dune may continue to migrate in its altered morphology (Fig. 1dii).

**Dune washout**. Continued and repeated occurrence of the pit formations demonstrated that the process of plastic particle erosion on the stoss side of a dune could result in dune wash out and formation of planar bed conditions[56] (Fig. 1d, 2ci-iii). The resulting erosion and dune morphologies inflicted by introduction of plastic in the channel causes a higher ratio of suspended to bedload sand locally, than would occur in an unpolluted setting (Fig. 1a). The repeated occurrences of stoss-side pit formation (Fig. 1cii) may amount to a more substantial set of consequences however, including instances of dunes being washed-out, a phenomenon that can happen under changing flood stage[56] (Figs. 1di; 1dii) but observed here under constant flow conditions exclusively due to the presence of plastic particles. The amount of plastic eroded from the stoss side of the dune dictates the extent of adjustment required by the dune to re-equilibrate with the flow. In all cases, the pits and scours migrating towards the dune crest reduce the dune volume, increase the downstream migration rate, and suggest that a local increase in the overall rate of downstream sand transport is likely (Figs. 1cii; 1dii). As the crest of the dune becomes more rounded or flattened, the angle of the lee slope may become too shallow to sufficiently shelter sediment for deposition in the lee side, or it may become entirely indistinct, such that the separation vortex in the recirculation zone disappears and the dune is washed out (Fig. 1dii). Such change to the riverbed topography was also found to affect dunes downstream of the disruption although this was not investigated in detail in this study. These changes would affect the overall resistance experienced by the flow, potentially changing the river stage locally; a flat riverbed offers less resistance to the flow than a riverbed with dunes and could have lower water depths. Further investigations, focused on quantifying flow depths and adaptations to changes in the riverbed due to the presence of plastic pollution, are needed to further elucidate the effects we describe.

Where the stoss side of the dune becomes markedly over steepened, a unidirectional symmetrical dune may occur (Fig. 1di). In this experiment, the rate of erosion on the stoss side of the dune is increased by plastic erosion, whilst the deposition on the lee side is slowed due to plastic movement in the recirculation zone, or the re-equilibrating of the dune. Such a scenario does not always lead to the dune becoming washed out, particularly if sufficient time passes between plastic erosion events allowing the dune to re-equilibrate (Fig 1dii). Alternatively, the dune remnants may remain and later be either reworked or overlain by the next dune migration, particularly if dune celerity is slow.

**Heterogeneous deposits**. Despite initially mixing the plastic particles into the sand, the plastic became organised and clustered during the experiment, and remained sufficiently abundant on the riverbed to cause multi-layered deposition throughout the experiment (Fig. 1bi). In the resulting deposit, plastic was found to be pervasive, yet there was a profound spatial variability in the plastic-to-sand ratio both vertically and laterally throughout the deposit (Fig. 3). Larger plastic particles (>4 mm) were found to be stored in the sediment as lenses, strings, and individual clasts (Fig. 1bii), whilst the smaller particles were found to have been distributed more generally, though not equally. Thermosetting plastic fragments showed a tendency to form distinguishable layers and lenses, that themselves may contain larger plastic particles (Fig. 3a). The thermosetting plastic fragments additionally highlighted preserved lee slopes, which mark the downstream progression of the dune (Fig. 3b) in cross-sets[27] (Fig. 1a). However, some sections of the deposit seem to be devoid of readily visible plastic particles (Fig. 3a), and such plastic-limited zones are found to be vertically and horizontally close to plastic-rich lenses (Fig. 3a, b). Additionally, where lenses from past dunes have not been entirely eroded, they may be overprinted by a new migrating dune, such that the overlying dune topography may not always align predictably with expected plastic-rich zones (Fig. 3c).

**Key findings**. Observing riverbed dynamics in the field is very difficult, particularly as the observations must be carried out without disturbing the occurring processes. To overcome this, fluvial geomorphologists have used small-scale experiments in the laboratory, for many decades, and it is widely accepted that these kind of experiments have an "unreasonable effectiveness"[50] due to natural scale independence, i.e. the underlying physics express themselves despite the difference in the spatio-temporal scales.

Paola et al.[50] is a review paper with many further articles cited throughout that may be found to also support the use of laboratory experiments representing the real world due to the well-documented and accepted morphodynamic similarity principle. Our experiments, like many others, are imperfect and limited, but the observations are likely consistent with field systems. Additionally, natural river systems are inherently complex, therefore flume tank experiments offer an excellent opportunity to isolate a variable, such that we can better understand that component of the complexity.

*Plastic is not a passive component of river systems*. Plastic is not a passively transported component of riverbeds; it interacts profoundly with sediment and plays an active role in affecting contemporary sedimentary systems during its transport. Plastic particles of different properties had different effects at this 0.5 m/s flow speed, for example, PVC powder was not observed to modify sand transport, yet the low mobility Mardi Gras beads and cigarette filter tips were consistently observed as impacting dune morphologies. How different proportions of plastic particles, perhaps of different densities, affect the flow at different speeds was beyond the remit of our experiments and will be addressed in future work. The plastic particles become included in, and affect, riverbed dunes under a constant flow velocity and at low concentrations, 0.12% by mass (1.49 kg) in our experiments; the plastic particles that we recorded regularly interacting with the bedform morphology are 0.081% by mass (1.00 kg), indicated with a star (*) on Table 1. The speed of bedform morphological transformation was found to increase with the addition of plastic, which affects bed topography and enhances dune erosion. This is critically important because it demonstrates for the first time that plastic can fundamentally change the local conditions and sediment transport mechanics at the riverbed, which may in turn alter its suitability for biotic habitation and impact longer-term river channel evolution, i.e., the observed flattening of riverbed topography lowers the hydraulic roughness (resistance to the flow offered by the riverbed), which could over time result in shallower rivers. On the other hand, if the disruptions to the bed resulted in shorter but taller bedforms, the hydraulic roughness would be higher and flood stage would increase. This aspect needs further investigation.

*Plastic inclusion locally changes the ratio of suspended load to bedload material*. The introduction of plastic was found to substantially disrupt local bedload sand migration, leading to smaller dunes of more variable size and morphological form. Plastic

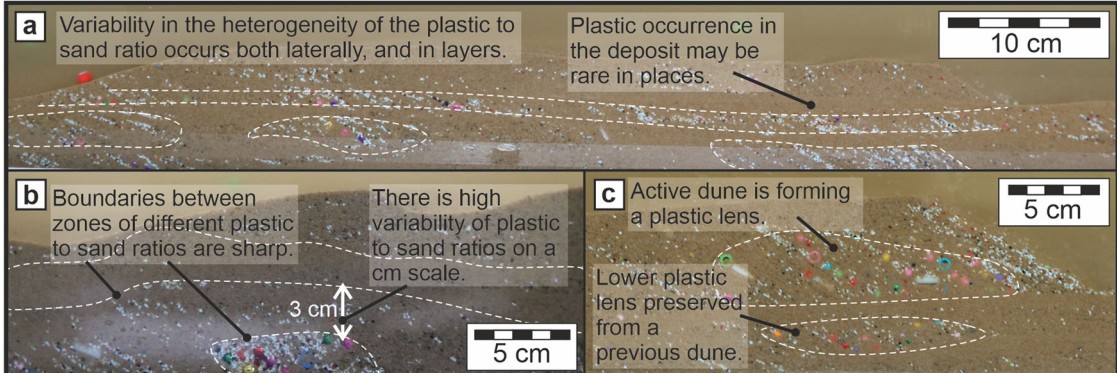

**Fig. 3 Images of the variability of the deposit resulting from the plastic and sediment mixture under constant flow.** Flow direction is left to right in all images and boundaries between different concentrations of plastic in sediment are demarcated with a white dotted line. **a** Layers and lenses of plastic-rich sediment exist throughout the deposit; **b** Sharp boundaries exist between high-, moderate-, and low-, plastic to sand ratio zones on a small scale; **c** Whilst the most recent plastic lenses may be found under the crest and in the body of the dune, older plastic-rich zones are less predictable to locate.

erosion causes a local and temporal shift towards more sediment in suspension with unknown consequences for overall sediment transport fluxes and increased local turbidity, potentially affecting light availability and the biotic processes that rely on it in the benthic zone. Additionally, suspended material tends to travel more rapidly than bedload material, and thus the local down-stream sand transport rates may increase under these conditions, though further studies are needed to confirm this both in the laboratory and in real-world rivers.

*Inclusion of plastic encourages rivers to locally develop more conduit-like properties and create heterogeneous deposits.* Within the experiments, sand and plastic are both stored in the riverbed and transported along it (Figs. 1 and 3). However, the physical properties of plastic disrupt sedimentary processes leading to a local increase of sand transported in suspension compared with a sand-only system, i.e., creating more conduit-like than storage-like properties[57]. The increase in material flux downstream has the potential to impact a wide range of fluvially influenced landscapes from mountains to the ocean, and could ultimately play a role in wider landscape evolution over longer time scales. The position of stored plastic in the sand bed was largely controlled by the properties of the plastic type, such as their shape, size, and density. Our results demonstrated an extremely heterogeneous distribution of plastic within sand beds – plastic may be concentrated in layers or isolated lenses or occur as individual particles (Fig. 3).

The plastic was all added to the flume tank at the beginning of the experiment, yet we saw the mechanics described in this manuscript continue for the whole observation period (>12 h). Therefore, we can conclude that even if plastic input ceases, the disruption of sand fluxes and dune migration processes may continue over time through the erosion, exhumation, and reworking of plastic-rich lenses already deposited (Figs. 1c, 1d, 2b, 2c, and 3). Critically, such spatial heterogeneity of plastic of different types in riverbed sediments renders quantifying plastic abundance in river sand a major challenge to representatively record.

**Challenges and opportunities.** Whilst the findings of this study offer a new perspective of riverbed processes that may be related to the natural world, there remains much to be done in understanding the composition, distribution, and abundance of plastics in riverbeds. Scaling from experimental flume experiments to nature is widely and successfully achieved[50], though we bear in mind that larger dunes in larger rivers may well be far less impacted by plastics of the scale used in this experiment as their dune troughs are much larger and deeper. Such scaling challenges and the combined impacts of plastic particle density, concentration, and distribution, in affecting thresholds for change is an emerging field of study with many outstanding queries.

The results of this study show that where sediment and plastic are found together, such as in rivers or drainage conduits, the

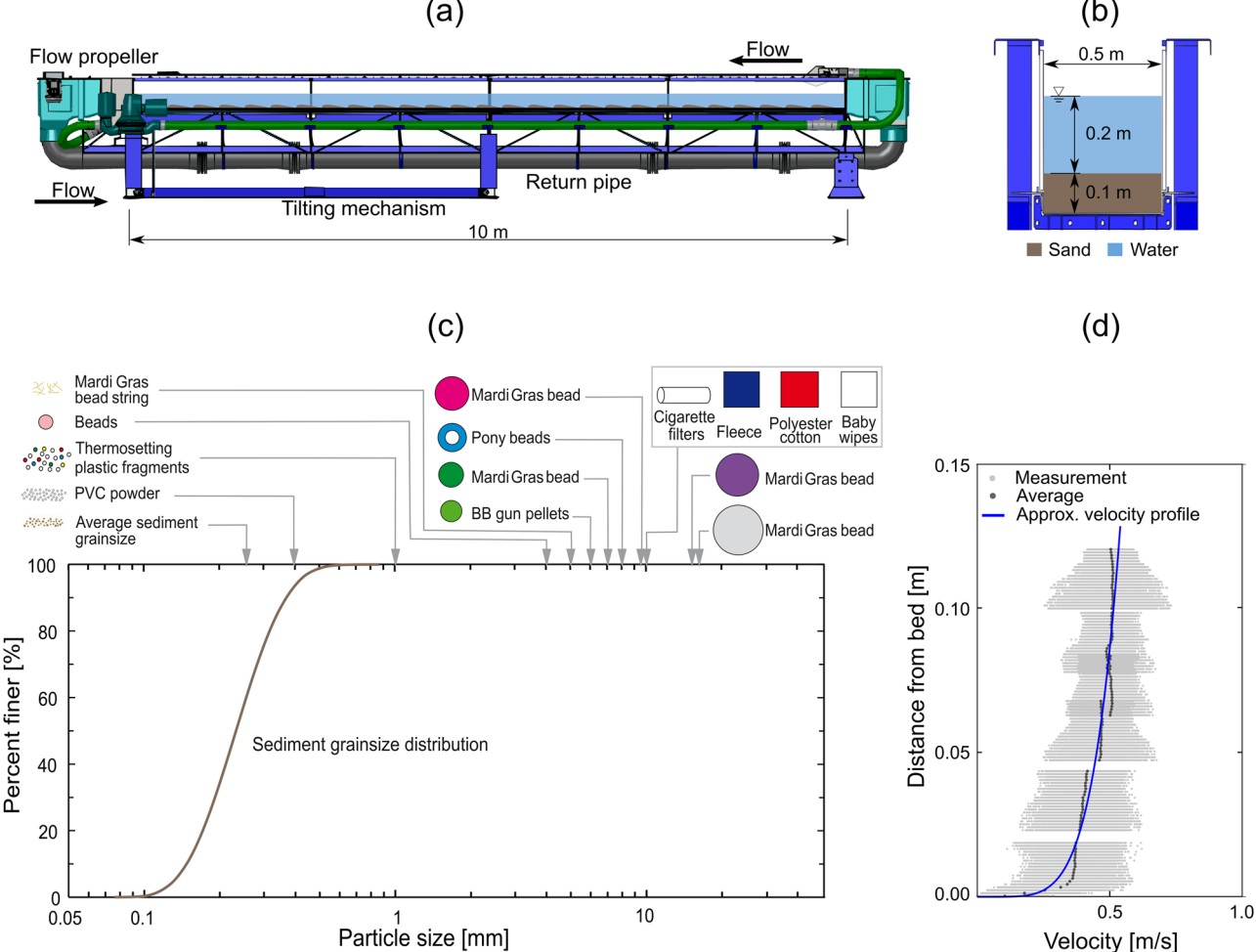

**Fig. 4 The flume tank, particles, and flow measurements used in the experiment. a** Side-view of the recirculating flume tank; **b** cross-section view of the recirculating flume tank; and **c** Sediment grain size distribution and plastic particle sizes; **d** flow velocity measurements and profile.

included plastic fundamentally affects the grain-to-grain mechanics of the sand transport. Additionally, when sampling plastic in riverbeds, from this study we now find that it should be from the crest of the dune to a depth that is concurrent with the dune height. Amending the sampling approach would allow for more accurate data on the abundance and composition of plastic litter in rivers, which in turn will allow for flume tank experiments to be more representative of contemporary conditions. Additionally, far more work is required to understand the down-stream effects of changes to these processes locally, such that we may quantify rates of change at a wider scale.

The impacts of plastic across Earth's environments remains a developing research field. The study herein outlines the first detailed process observations of plastic interacting with sandy bedforms on a riverbed, and sheds light into a new branch of environmental monitoring and sedimentology. It is critical that we continue to explore these novel dynamics with more laboratory experiments, field monitoring, and computer models to refine our understanding of these newly established processes. As this new chapter of sedimentology emerges, we are able explore further the relationship of plastic with sediment across our landscapes and its long-term consequences.

## Methods

We conducted a 26-h experiment in a 10 m long and 0.5 m wide recirculating flume at the University of Hull, UK (Fig. 4a, b). The flume has a maximum depth of 0.5 m and we filled the bottom 0.1 m with sand with a median sediment size $D_{50}$ of 0.23 mm (Fig. 4c). We filled the flume with water so that the flow depth above the, initially, flat and level sand bed was 0.2 m. A constant flow discharge of 0.05 m³/s was maintained throughout the experiments, giving a depth-averaged flow velocity of 0.5 m/s (Fig. 4d). Additional relevant parameters associated with these flow conditions and a water temperature of 18 °C are: Froude number – Fr = 0.36; Reynolds number – Re ~ 92,600; Bed shear stress after correcting for sidewall effects[58,59] – $\tau_{bed}$ = 1.3 Pa; Shields number – $\tau^*$ = 0.35. As this is an emerging discipline in understanding how plastic and sand interact on a riverbed, all insights would be novel and of value, so these conditions were selected because they produced dunes of an appropriate scale for the pollutants, and the flow speed was rapid enough to transport the particles.

After the flume had been running for 12 h and the bed had attained equilibrium conditions with the flow, a detailed velocity profile was constructed from measurements over a migrating train of dunes within the central test section of the flume (at 5 to 7 m from the flume flow entry; Fig. 4) using an Acoustic Doppler Velocimeter (ADV)[60].

After characterizing the flow, a 1.49 kg mixture with 13 different types of plastics, including particles, fibres, and fragments, were introduced into the flume (Table 1). The mass of plastic added to the flume only represents 0.12%, by mass, of the total sediment used within the flume system.

Directly observing sediment dynamics within a natural riverbed is a substantial challenge, meaning that ground-truthing experimental results in the field is complex. A range of approaches have been attempted such as Morritt et al.[45] where nets were anchored to the bottom of a river. However, it was only possible to position these nets ~40 cm above the riverbed, meaning the interactions of plastic with the bed remained uncertain and largely unquantified. Herein we used a range of generic plastic types found in river systems, including both elongate (cigarette filter tips) and rounded shapes of different sizes and densities (Eq. 1; Table 1). Rounded shapes readily represent the endemic challenge of nurdle spills[61], which although they are most studied in marine settings, also occur along rivers, e.g., the Mississippi River. Additionally, rounded shapes are an ideal and readily relatable starting point for translating existing knowledge of sedimentary mechanics to find the knowledge gaps for further study.

We recognise that the scale of the plastics in relation to the size of the morphological unit scale is important in the experimental design. Whilst the water depth and dune size may be small compared to the larger rivers found in nature, these processes may occur anywhere that sediment and plastics coexist, such as drainage routes or small urban streams, which tend to be the most polluting[12]. All observations associated with sediment transport mechanics and their disruption remain valid even if the changes to dune morphologies are only applicable to small dunes. Moreover, small-scale laboratory experiments where sediment transport processes are the main focus, have proven to be quite useful for the geomorphology and sedimentology communities[50].

During the experiments, different particles showed different behaviours in terms of mobility under the flow conditions used. The last column in Table 1 shows our observations. To identify the threshold between high/low mobility we computed the product of R, the particle's submerged specific gravity (Eq. 1), and its equivalent diameter D and took the ratio with respect to the sand grains used in the

experiment. The next-to-last column in Table 1 shows how much more mobile the corresponding plastic particle is with respect to the sand grains used in the experiment. The threshold between high/low mobility for the flow conditions used lies at approximately RD ~ 1.

The submerged specific gravity is computed as:

$$R = \frac{\rho_p}{\rho} - 1 \qquad (1)$$

Where $\rho_p$ is the density of the particle (see Table 1 for values) and $\rho$ is the density of the fluid (water in this case).

Equivalent particle diameters for non-spherical particles were computed as:

$$D = \sqrt[3]{\frac{6V_p}{\pi}} \qquad (2)$$

Where $V_p$ is the particle volume. For the fleece, polyester, and baby wipe fragments, a thickness of 0.1 mm was assumed.

Figure 4c demonstrates the sizes and types of the different particles used in the experiment. Plastics were left to be transported by the flow for 120 min, after which the flow discharge was increased up to 0.06 m³/s to simulate flood conditions and promote additional mixing of plastics with the sandy bed. After a period of 120 min the flow was returned to 0.05 m³/s and observations were made of the plastic transport and the bed interactions. This included video capture (see Supplementary Movie 1) and detailed mapping of the transport pathways.

## Data availability

The data that support the findings of this study are openly available in this article and its Supplementary Materials. Flow velocity data used to construct Fig. 4d is available under Creative Commons Attribution 4.0 International (Russell et al.[60] https://doi.org/10.5281/zenodo.7783622).

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

## Acknowledgements

The authors are thankful for the help offered by Connor Burchell, Marijke de Vet, Stuart McLelland, Greg Smith, and Clint Wilson during different stages of our experiments, including setup, measurements, and clean up. Participation of RF was made possible thanks to funding from the European Research Council under the European Union's Horizon 2020 research and innovation program (Grant 725955) and UK NERC funding (NE/X018091/1), and the Leverhulme Trust, Leverhulme Early Career Researcher Fellowship (grant ECF-2020-679).

## Author contributions

C.R. conceived and led the project. C.R., R.F. and S.E.G. designed the experiments and methodology. C.R. and R.F. performed the experiments and analysed the data with contributions by D.P. C.R. wrote the first manuscript draft and contributed Figs. 1–3 and the Supplementary Materials, and RF wrote the Methods and contributed Fig. 4. All authors discussed results and implications and commented on the manuscript at all stages.

**Competing interests**
The authors declare no competing interests.

**Ethics approval**
Any waste arisings were disposed of with a licensed waste disposal operator.

**Additional information**

