## [Peer Review File · Communications Earth & Environment]

17th Jun 22

Dear Dr Russell,

Please allow me to apologise for the long delay in sending a decision on your manuscript titled "Plastic pollution in riverbeds fundamentally affects natural sand transport". It has now been seen by 2 reviewers, whose comments are appended below. You will see that they find your work of some potential interest. However, they have raised quite substantial concerns that must be addressed. In light of these comments, we cannot accept the manuscript for publication, but would be interested in considering a revised version that fully addresses these serious concerns.

In particular, we will need to see that any revised manuscript meets the following editorial thresholds:

** Fully discuss the limitations of your findings and experiments in the context of sediment dynamics in natural river systems and adapt and soften your conclusions accordingly **

** Provide sufficient detail of your experimental parameters and experimental approach to ensure your results are reproducible **

We hope you will find the reviewers' comments useful as you decide how to proceed. Should additional work allow you to address these criticisms, we would be happy to look at a substantially revised manuscript. If you choose to take up this option, please either highlight all changes in the manuscript text file, or provide a list of the changes to the manuscript with your responses to the reviewers.

Please note that although Reviewer #1 requests that you move the Methods to immediately after the Introduction, it is our journal style to include the Methods at the end of the paper, after the Results and Discussion. That said, they do need to be moved ahead of the References and you can include a short summary of the Methods and experimental approach at the end of the Introduction or at the start of the Results section. Please also note that although Reviewer #2 suggests the removal of some basic background information, we do cater to an audience with a wide range of specialisms so the level of this information you choose to provide is at your discretion.

If the revision process takes significantly longer than three months, we will be happy to reconsider your paper at a later date, as long as nothing similar has been accepted for publication at Communications Earth & Environment or published elsewhere in the meantime.

We understand that due to the current global situation, the time required for revision may be longer than usual. We would appreciate it if you could keep us informed about an estimated timescale for resubmission, to facilitate our planning. Of course, if you are unable to

estimate, we are happy to accommodate necessary extensions nevertheless.

Please use the following link to submit your revised manuscript, point-by-point response to the reviewers' comments with a list of your changes to the manuscript text (which should be in a separate document to any cover letter) and any completed checklist:

[link redacted]

Please do not hesitate to contact me if you have any questions or would like to discuss the required revisions further. Thank you for the opportunity to review your work.

Best regards,

Joe Aslin

Locum Chief Editor,
Communications Earth & Environment
<https://www.nature.com/commsenv/>
Twitter: @CommsEarth

EDITORIAL POLICIES AND FORMAT

If you decide to resubmit your paper, please ensure that your manuscript complies with our editorial policies and complete and upload the checklist below as a Related Manuscript file type with the revised article:

Editorial Policy <[a href="https://www.nature.com/documents/nr-editorial-policy-checklist.zip"](https://www.nature.com/documents/nr-editorial-policy-checklist.zip)>Policy requirements

For your information, you can find some guidance regarding format requirements summarized on the following checklist:(<https://www.nature.com/documents/commsj-phys-style-formatting-checklist-article.pdf>) and formatting guide (<https://www.nature.com/documents/commsj-phys-style-formatting-guide-accept.pdf>).

REVIEWER COMMENTS:

Reviewer #1 (Remarks to the Author):

The presence of plastic pollution in river ecosystems dramatically increased in the last decades. The paper deals with an emergent topic which gained lots of interest from the scientific community, including a very wide field of applications and base research. In particular, we recognized that plastic particles undergo transport process in some way similar to sediment particles, but with several important differences which attain to shape, density and roundness of plastic particles.

The experiment is sufficiently described, the video provided in the Supplementary Materiale are very nice and help to understand the processes. Some details of the experiment need to be better described. After my review of the paper, my recommendation is "Accept with Minor revision".

Some suggestions follow.

1. The introduction is missing on the discussion of previous works regarding transport processes experienced by plastic particles. The introduction will benefit from the addition of some of the recent works on plastic transport, which investigated settling velocity (see among the others Khatmullina and Isachenko (2017), Waldschlager and Schüttrumpf (2019a), Francalanci et al. (2021)), the influence of shape and density in transport processes, and the influence of the threshold of incipient motion (see for example Waldschlager and Schüttrumpf, 2019b), but there are several others references that can be added.
2. Method section. I do not understand why the "method" section is after the references. At the beginning I thought it was an intentional choice, but after my review, and given the short length of this section, I believe it was a mistake with the document. I recommend moving the method section after the introduction section. The information reported in the Method section are essential to the reader for understanding the results and the discussion, they are not optional, so the paper needs to be re-organized.
3. Results and Discussion. Lines 101-112. Some details of the experiments are anticipated here, but this short description is not sufficient for the reader to have a minimum understanding of what the experiments is about: how much is the percentual of plastic in the sand? How many experiments did you run? Which are the characteristics of plastic particles used in the experiment? The reader is forced to skip the text and look at the end of the paper for this information, in order to understand the results and discussion section. These additional information should be added to the experiment description. Moreover, Table 1 is cited here (Line 106, 112) but is much farther in the text. My suggestion is a better structure of the paper, reorganizing the sections, Figures and Table.
4. The presentation of the results is very clear and informative. I suggest citing the Video Clip number when discussing the observed processes. The Supplementary Material is very important to understand the processes described in the paper.
5. Lines 454 and following. How is the plastic mixture composed? The total weight is not enough to reproduce the experiment. The authors correctly said that "the scale of the plastics in relation to the size of the morphological unit scale is important in the experimental design", so how do you scale the size, density and quantity of plastic observed in the urban stream in Meijer et al., 2021?
6. Moreover, if I understood correctly the authors carried out only one experiment: I am not sure that with only one experiment they can address general conclusion; they can describe what they observed but a relevant number of questions remain unresolved: which is the influence of the composition of the plastic mixture added to the sand? Do you try to repeat

the experiment to verify the results? Can you expect similar results if you change plastic percentual and plastic composition?

Minor points.

Figure 4. Move the Arrow “Flow” on the left to a better spot; now it seems connected in some way with the tilting mechanism. It would be better to move it below the recirculating pipe.

Lines 451-452 – Can you add the location of the test section in the flume in panel a) Figure 4?

Lines 463-467. It took me a while to understand that RD is R multiplied by D (diameter). You can rephrase the sentence, for example: “to identify the threshold We computed the parameter RD as the product of R, particle’s submerged , and its equivalent diameter D ...” In the table change “Ratio to sand (-)” with “RD ratio to sand (-)”

Table 1 and Table 2 are the same.

References

- Khatmullina, L., Isachenko, I., 2017. Settling velocity of microplastic particles of regular shapes. *Mar. Pollut. Bull.* 114, 871–880. <https://doi.org/10.1016/j.marpolbul.2016.11.024>.
- Waldschlager, K., Schüttrumpf, H., 2019. Effects of particle properties on the settling and rise velocities of microplastics in freshwater under laboratory conditions. *Environ. Sci. Technol.* (53). <https://doi.org/10.1021/acs.est.8b06794>, 1958–1966.
- Waldschlager, K., Schüttrumpf, H., 2019. Erosion Behavior of Different Microplastic Particles in Comparison to Natural Sediments. *Environ. Sci. Technol.* 2019, 53, 13219–13227. <https://doi.org/10.1021/acs.est.9b05394>.
- Francalanci, S., Paris, E., Solari, L. (2021). On the prediction of settling velocity for plastic particles of different shapes. *Environmental Pollution*, 2021, 290, 118068. <https://doi.org/10.1016/j.envpol.2021.118068>

Reviewer #2 (Remarks to the Author):

COMMSENV-22-0233-T Plastic pollution in riverbeds fundamentally affects natural sand transport

Comments by J.H. (Janrik) van den Berg

28-5-2022

This might become a valuable contribution as it contains novel information on the influence of plastic particles on dune dynamics. However, its far-reaching claims of its value to nature are not justified by the experiments executed, because they do not reflect conditions found in many natural rivers. I would therefore advise the authors to restrict their conclusions to the results of the experiments and to be humble and very reserved about their generic value. A lot of the text, especially in the Introduction, is not essential to what should be the main objective of the paper: description and analysis of “the effect of a mixture of plastic particles on bedform dynamics in some flume experiments” (“...” suitable title of the paper) and

therefore can – and should - be omitted. Conversely, text and figures should be added to (1) document better the experiments (2) integrate information on the effect of natural low-density admixtures (organic debris) on dune dynamics. (3) discuss to what extent the experiments represent natural conditions and to adapt the conclusions accordingly.

In a flume with a water depth of two dm only small dunes can develop, much smaller than usually found in a river. In fact, the bedforms studied are only up to about 5 cm high, which is only slightly higher than the size of ripples and smaller than any natural dune found developed in conditions where size is not limited by water depth. With larger, more natural dunes, accumulations of plastic in the dune trough and its effect on the dune shape may have a much smaller effect as observed in the experiments, because those dune troughs are much larger and deeper. An indication of this is the influence of organic detritus on crossbedding as recorded in some previous literature. I would recommend going into this and pointing out the similarities/differences with the interaction by plastic debris. Some examples can be found in the book I wrote with Allard Martinius “Atlas of sedimentary structures in estuarine and tidally-influenced river deposits of the Holocene Rhine-Meuse-Scheldt system” (e.g., page 107, figure 3.6.1; page 128, figures 4.1.9 and 4.1.10). Organic material is deposited in the toe of the dunes during periods of low flow speed (see also Martinius and Gowland, 2010: <https://doi.org/10.1111/j.1365-3091.2010.01185.x>). This raises the question under which experimental conditions of dune formation the accumulations of the plastic particles occurred. In other words, where do the experiments plot in stability diagrams, just above the start of dune formation?

Another concern in these experiments is to what extent the material choice and concentration approach reality. Is the mixture representative of conditions in nature in terms of particle shape and grain size distribution? To be able to test this in future research, it is necessary that the authors give grain distributions of the mixtures used. It is also necessary to provide a useful measure for the concentration, 0.12% by mass of the sand present in the flume is not. I propose the volume percentage in relation to the sand in the active layer (= dune volume) per meter of stream length and width.

Additional comments I have inserted in the margin of the MS transferred to Word.

**Plastic pollution in riverbeds fundamentally affects natural sand transport**

Catherine E. Russell^{1*}, Roberto Fernández², Daniel R. Parsons², and Sarah E. Gabbott¹

¹School of Geography, Geology, and the Environment, University of Leicester, UK, LE1 7RH

²Energy and Environment Institute, University of Hull, UK, HU6 7RX

*Corresponding author – cr295@le.ac.uk (ORCID - 0000-0002-6842-6329)

**Rivers link terrestrial and ocean environments, distributing fresh water, nutrients, and**
**sediment to diverse ecosystems. Over the past 50 years, rivers have become increasingly**
**significant vectors for plastic pollution. Lowland riverbeds exhibit coherent features**
**including ripple and dune bedforms, which transport sediment downstream via well-**
**understood processes, yet the impact of plastic on sediment transport behaviours is**
**largely unknown. Here we use a flume tank to show that when plastic particles are**
**introduced to sandy riverbeds, even at relatively low concentrations, novel bedform**
**morphologies and altered processes emerge, including irregular bedform stoss erosion**
**and dune “washout”, causing topographic bedform amplitudes to decline. We detail i)**
**new mechanisms of plastic incorporation and transport in riverbed dunes, and ii) how**
**sedimentary processes are fundamentally influenced. Hence, plastic is not a passive**
**component of river systems; it directly affects bed topography, whilst increasing the**
**proportion of sand suspended in the water, which has the potential to impact river**
**ecosystems and the wider landscape. The resulting plastic distribution in the sediment is**
**heterogeneous, highlighting the challenge of representatively sampling plastic**
**concentrations. Our insights establish a new branch of process sedimentology: plastic and**
**sand interactions, set to be increasingly relevant amongst emerging challenges of the**
**Anthropocene.**

**Introduction**

Plastics, and their derived products, are pervasive within earth surface systems globally; they
are in the air that we breathe (Gasperi et al., 2018; Bullard et al., 2021), in agricultural soils
(Liu et al., 2018; Corradini et al., 2019), in aquatic biota (Rezania et al., 2018; Zhu et al., 2019),
throughout fresh and marine waterbodies (Lee et al., 2015; Hurley et al., 2018; Pohl et al.,
2020), and in the deepest abyssal trenches (Peng et al., 2020; Kane et al., 2020). Rivers are the

Commented [A1]: Title incorrect, because sand transport was not measured! Suggestion for title that better fits the subject: “Influence of plastic particles on dune dynamics in some flume experiments”

Commented [A2]: Omit, because not essential to the subject

Commented [A3]: This far-reaching conclusion cannot be drawn from the experiments!

primary terrestrial conduits of plastic, carrying and delivering an estimated 12 M metric tonnes
of plastics to coastal and marine environments each year, with this number set to at least double
by 2030 (Jambeck et al. 2015). Yet, despite advancements in understanding how plastic travels
in rivers (van Emmerik and Schwarz, 2019; Liro et al., 2020; Tasserson et al., 2020), our
understanding of its impact on broader sediment transport processes and sedimentary systems
more generally is in its infancy (Gabbott et al., 2020), which has implications for environmental
monitoring, representative sampling, landscape evolution and sedimentary geology.

Plastic has a wide range of physical properties, morphologies, and characteristics (GESAMP,
2015), such that particles less dense than fresh water will travel floating on or near the water
surface, whereas denser particles will travel closer to the riverbed, or even bounce or roll along
it (Morritt et al., 2014). A significant volume of plastic manufactured is denser than freshwater
(Lebreton et al., 2019), resulting in large quantities of plastic travelling under the surface waters
of rivers (Morritt et al., 2014; Al-Zawaidah et al. 2021) and interacting with, or being passively
stored in, the riverbed (Ockelford et al., 2020; Liro et al., 2020; van Emmerik et al., 2022).
Observing and sampling riverbed processes in natural environments without disturbing
them is challenging, and thus monitoring and characterising the effect of plastic on
riverbed transport processes is difficult. Therefore, we designed and undertook physical
laboratory experiments (see Methods) to explore whether plastic influences river transport
mechanisms and focused on representing small urban stream systems, which are
commonly highly contaminated with plastic due to their proximity to pollution sources
(Meijer et al., 2021).

Despite a high variability in river shape globally (Church, 2006; Rinaldi et al. 2016), dunes
are ubiquitous features on their mobile riverbeds (Venditti et al., 2005; Ohata et al., 2017;
Cisneros et al., 2020). River dunes, which have a shallow upstream stoss side, and a steeper
downstream lee-side, migrate downstream by the continuous erosion of sediment particles
from the stoss-side and subsequent deposition of material on the lee side (Kennedy, 1963;
Allen, 1982) (Fig. 1A). The downstream migration of these riverbed dunes influences the
shape of the river channel (Unsworth et al., 2020), and they are important in controlling and
maintaining sediment flux that is crucial for a suite of ecosystem services (e.g. McCulloch et
al., 2003; Ludwig et al., 2009). Bedform geometries and processes are dictated by the grain-to-
grain interactions between the sediment particles, yet, we know biota can influence such small-
scale grain-to-grain interactions, which can compound to result in large-scale and significant

Commented [A4]: You cannot extent conclusions from experimental results to nature if you don't know whether experimental conditions approach nature.

Commented [A5]: Well-known to everybody, can be omitted

wide-reaching effects on landscape dynamics (Dietrich and Perron, 2006; Polvi and Wohl,
2013; Fremier et al., 2018). Biotic processes alter and interact with bedform dynamics
(Gottesfeld et al., 2004; Statzner, 2012; Parsons et al., 2016), altering topographies (Hassan et
al., 2008; Fremier et al., 2018; Han et al., 2019), grain size distributions (Rice et al., 2019), and
particle cohesion (Grant et al., 1986; Statzner, 2012). Plastic is found on riverbeds and is
characterised by more variable properties than natural sediment, therefore, the fundamental
grain-to-grain interactions that underpin bedform dynamics, sediment transport processes, and
the wider landscape may be disrupted by its presence. However, the impact of plastic on
riverbed dynamics is yet to be investigated and is the focus of our experiments.

Commented [A6]: Not really relevant to this paper, can be omitted

Through contaminating sand dunes with plastic in a recirculating laboratory flume tank, we
test how the inclusion of plastic particles in riverbed sand dunes impacts morphologies and
migration processes. We explore how and where plastics interact with bedform dynamics, and
how these interactions impact a host of processes known to be important for sediment transport
and hydrology, as well as the resulting sedimentary deposit. We seek to understand what the
results mean for the fate and distribution of plastic and demonstrate important implications for
representative sampling of plastic in riverbeds, as well as understanding the controls on where
plastics may be concentrated in certain environments. We advocate that our results offer
fundamental insights on what we suggest is a new sub-branch of sedimentology: sediment and
plastic particle interactions, set to be increasingly relevant as earth scientists prepare to
describe, interpret, and understand Anthropocene landscapes.

Commented [A7]: This is the basic information needed. In fact all other information given in this Introduction is not adding much to this and therefore in my opinion can be left out.

Results and Discussion

River dunes typically migrate by progressive erosion of sand grains from the stoss side and
deposition on the lee side (Fig. 1A). Overall, flow patterns in river channels are affected by the
presence of dunes, with higher flow velocities above the dune crest and slower flow velocities
above the dune trough (e.g. Bennett and Best 1995, Fig. 1Ai). The dune shape itself often leads
to flow separation (detachment) and a recirculation zone within the dune lee side, where flow
velocities reduce, and sediment grains tend to be deposited (e.g. Reesink et al. 2018). The
results herein show that these well-established flow and sediment transport processes are
impacted by the presence of plastic particles, that often have lower density than sand and
therefore are easier to entrain and transport.

Commented [A8]: This is basic info, known by everybody, can be left out

Commented [A9]: ...the well-established flow and sediment transport processes over dunes...

Sand (median sediment size D_{50} of 0.35 mm) and 13 different types of plastic were put into a
recirculating laboratory flume tank under a constant flow discharge (of 0.05 m³/s) to observe
interactions between the plastic and sand under transport conditions (see Methods for full
details). The size and density of the plastic and the sand defines their mobility, thereby their
erosion and entrainment thresholds. The plastics used in this experiment had a lower density
than sand (Table 1), however, some had larger sizes and therefore comparatively lower
mobility, leading to a range of complex interactions. The results indicate that, for the flow
conditions in the flume tank and the particles used, their behaviour and interaction with the
sand bed changed at $RD \sim 1.6$, (where RD is the particle diameter multiplied by submerged
specific gravity). Particles with $RD > 1.6$ were found to be less mobile and more readily
incorporated into the sand dune, than particles with $RD < 1.6$, which had greater mobility and
were less readily incorporated into the sand substrate (Table 1).

Commented [A10]: D_{50} and D_{90} of 0.35 mm and ... mm resp., see also Fig. ... (=grainsize distribution = NEW FIG.)

Commented [A11]: Provide graph of grainsize distribution

Commented [A12]: This is not a useful parameter. Provide information such as average values of flow, bed shear stress, Shields parameter etc and show how the experiments plot in well-known stability diagrams (do not forget to correct values for flume side-wall roughness).

*Figure 1 – Processes by which sand is included in, and subsequently eroded from, sand dunes.*
*A: The standard terminologies and classic dune migration process; B: Lee-side processes in a*
*plastic polluted sand dune; C: Stoss-side processes in a plastic polluted sand dune and*
*consequential eroded profiles; D: Processes of dune wash-out whereby the extent of*
*disturbance of the dune by the plastic is too great for the dune to sustain its downstream*
*migration.*

*Lee-side Processes*

Migrating sand dunes were shown to trap plastics that accumulated within the lee side
[revised manuscript text omitted]
 rate of downstream sand transport will increase due to the reduction in dune volume**
**as the pits and scours migrate towards the dune crest (Figs 1Cii; 1Dii). As the crest of the dune**
**becomes more rounded or flattened, the angle of the lee slope may become too shallow to**
**sufficiently shelter sediment for deposition in the lee side, or become entirely indistinct, such**
**that the dune is washed out (Fig. 1Dii). Such change to the riverbed topography was also found**
**to affect dunes downstream of the disruption although this was not investigated in detail**
**herein.**

Commented [A14]: A reduction in dune volume results in a higher migration rate, not necessarily in a higher sediment transport.

Commented [A15]: If true in nature this would have two important consequences you might address:
1. With the hydraulic roughness would become much lower, resulting in less deep rivers (with possible consequence for navigation at low discharge and (less) flooding at peak discharge. Before making this conclusion generic, please check have such changes been recorded in rivers with plastic pollution?
2. when the separation vortex disappears, the hydraulic form roughness also disappears, leaving more energy for sand transport.

 *Figure 2 – Images of dunes from the experiment showing: A: Multi-layered deposition; B: A*
 *dune that has undertaken group erosion on the stoss-side and is now equilibrating; C: The*
 *erosion of filter tips at Ci and Cii leads to distinctly changed dune morphology.*

 Where the stoss side of the dune becomes significantly over steepened, a unidirectional
 symmetrical dune may occur (Fig. 1Di). In this experiment, the rate of erosion on the stoss side
 of the dune is increased by plastic removal, whilst the deposition on the lee side is slowed due
 to plastic movement in the recirculation zone, or the re-equilibrating of the dune. Such a
 scenario does not always lead to the dune becoming washed out, particularly if sufficient time
 passes between plastic erosion events allowing the dune to re-equilibrate (Fig 1Dii).
 Alternatively, the dune remnants may remain and later be either reworked or overlain by the
 next dune migration, particularly if dune celerity is slow.

 *Heterogeneous Deposits*

Despite the initial mixing of the plastic into the sand, the plastic became organised and clustered
 during the experiment, and remained sufficiently abundant on the riverbed to cause multi-
 layered deposition throughout the experiment (Fig. 1Bi). In the resulting deposit, plastic was
 found to be pervasive, yet there was a profound spatial variability in the plastic to sand ratio
 both vertically and laterally throughout the deposit (Fig. 3). Larger plastic particles (> 4 mm)
 were found to be stored in the sediment as lenses, strings, and individual clasts (Fig. 1Bii),

whilst the smaller particles were found to have been distributed more generally, though not
 equally. Thermosetting plastic fragments showed a tendency to form distinguishable layers and
 lenses, that themselves may contain larger plastic particles (Fig. 3A). The thermosetting plastic
 fragments additionally highlighted preserved lee slopes, which mark the downstream
 progression of the dune (Fig. 3B) in cross-sets (Allen, 1982; Fig. 1A). However, some sections
 of the deposit seem to be devoid of readily visible plastic particles (Fig. 3A), and such plastic-
 limited zones are found to be vertically and horizontally close to plastic-rich lenses (Figs 3A;
 3B). Additionally, where lenses from past dunes have not been entirely eroded, they may be
 overprinted by a new migrating dune, such that the overlying dune topography may not align
 predictably with expected plastic-rich zones (Fig. 3C).

*Figure 3 – Images to show the variability of the resulting deposit from the plastic and sediment*
 *mixture. Flow direction is left to right in all images. A: Layers and lenses of plastic-rich*
 *sediment exist throughout the deposit; B: Sharp boundaries exist between high-, moderate-,*
 *and low-, plastic to sand ratio zones on a small scale; C: Whilst the most recent plastic lenses*
 *may be found under the crest and in the body of the dune, older plastic-rich zones are less*
 *predictable to locate.*

**Key findings:**

1) Plastic is not a passive component of river systems

Plastic is not a passively transported component of riverbeds. It interacts profoundly with
 sediment and plays an active role in controlling contemporary sedimentary systems during its
 transport, particularly where concentrations approach similar levels to those investigated
 herein. Plastic particles become included in riverbed dunes and, even under a constant flow
 velocity and at low concentrations (0.12% by mass in our experiments), the speed of bedform
 morphological transformation was found to increase, which affects bed topography and

Commented [A16]: Restrict your findings to your experimental results as you have no any “ground truth” with natural rivers

enhances dune erosion rates. This is critically important because it demonstrates for the first
time that plastic can fundamentally change the conditions and sediment transport processes at
the riverbed, which will in turn alter its suitability for biotic habitation and impact longer-term
river channel evolution.

Commented [A17]: The experiments only show a reduction of dune volume, not a change in sediment transport

2) Plastic inclusion changes the ratio of suspended load to bedload material

The introduction of plastic was found to significantly disrupt bedload sand migration, leading
to smaller dunes of more variable size and morphological form. Plastics create a local and
temporal shift towards more sediment in suspension with unknown consequences for overall
sediment transport fluxes and increased local turbidity, potentially affecting light availability
and the biotic processes that rely on it in the benthic zone. Additionally, suspended material
tends to travel more rapidly than bedload material, and thus the downstream sand transport
rates will increase under these conditions.

Commented [A18]: You have not done any measurement of bedload transport!

Commented [A19]: You have not made any measurement of suspended bed-material transport!

3) Inclusion of plastic encourages rivers to develop more conduit-like properties and create 275 heterogeneous deposits.

Within the experiments, sand and plastic are both stored in the riverbed and transported along
it (Figs 1 and 3). However, the physical properties of plastic disrupt sedimentary processes
leading to a local increase of sand transported in suspension compared with a sand-only system,
i.e., creating more conduit-like than storage-like properties (Syvitski et al., 2022). This increase
in material flux downstream has the potential to impact a wide range of fluvially influenced
landscapes from mountains to the ocean, and ultimately could play a role in wider landscape
evolution over longer time scales. The position of stored plastic in the sand bed was largely
controlled by the properties of the plastic type, such as their shape, size, and density. Our results
demonstrated an extremely heterogeneous distribution of plastic within sand beds – plastic may
be concentrated in layers or isolated lenses or occur as individual particles (Fig. 3). The
incorporated plastic within sand suggests that even if plastic input ceases, the disruption of
sand fluxes and dune migration processes may continue over time through the erosion,
exhumation, and reworking of plastic-rich lenses already deposited. Critically, such spatial
heterogeneity of plastic of different types in riverbed sediments renders quantifying plastic
abundance in river sand a significant challenge to representatively record.

Commented [A20]: The introduction of plastic particles in a flow might damp turbulence which would counteract the increase of sand transport you suggest. So, in absence of any support from measurements your claim of increased sand transport is speculation.

Commented [A21]: This may be true, but not based on your experiments, and therefore should not be part of this chapter

The impacts of plastic across Earth's environments remains a developing research field. The
study herein outlines the first detailed process observations of plastic interacting with sandy

Commented [A22]: May be correct, but not based on data and rather far beyond your experiments: erase

bedforms on a riverbed, and sheds light into a new branch of environmental monitoring and
sedimentology. It is critical that we continue to explore these novel dynamics with more
laboratory experiments, field monitoring and computer models to refine our understanding of
these newly established processes. As this new chapter of sedimentology emerges, we are
able explore further the relationship of plastic with sediment across our landscapes and its
long-term consequences.

**References**

Allen, J.R.L., 1982. Sedimentary structures: Their Character and Physical Basis, Volume 1.
Developments in Sedimentology 30. Elsevier Science Publishers, **30**, pp.285-94.

Al-Zawaidah, H., Ravazzolo, D. and Friedrich, H., 2021. Macroplastics in rivers: present
knowledge, issues and challenges. *Environmental Science: Processes & Impacts*, **23**, pp.535-
552.

Bennett, S.J. and Best, J.L., 1995. Mean flow and turbulence structure over fixed, two-
dimensional dunes: implications for sediment transport and bedform stability. *Sedimentology*,
**42**, pp.491-513.

Bullard, J.E., Ockelford, A., O'Brien, P. and Neuman, C.M., 2021. Preferential transport of
microplastics by wind. *Atmospheric Environment*, **245**, pp.1-9.

Carling, P.A., Golz, E., Orr, H.G. and Radecki-Pawlik, A., 2000. The morphodynamics of
fluvial sand dunes in the River Rhine, near Mainz, Germany. I. Sedimentology and
morphology. *Sedimentology*, **47**, pp.227-252.

Church, M. (2006) Bed material transport and the morphology of alluvial river channels.
*Annual Review of Earth and Planetary Sciences*, **34**, 325-354.

320

Commented [A23]: I don't think the results of your interesting experiments are of such importance that they allow the recommendation to open a new field of science

- Cisneros, J., Best, J., van Dijk, T., de Almeida, R.P., Amsler, M., Boldt, J., Freitas, B., Galeazzi, C., Huizinga, R., Ianniruberto, M. and Ma, H., 2020. Dunes in the world's big rivers are characterized by low-angle lee-side slopes and a complex shape. *Nature Geoscience*, **13**, pp.156-162.

Corradini, F., Meza, P., Eguiluz, R., Casado, F., Huerta-Lwanga, E. and Geissen, V., 2019.
Evidence of microplastic accumulation in agricultural soils from sewage sludge disposal.
*Science of the total environment*, **671**, pp.411-420.

Dietrich, W.E. and Perron, J.T., 2006. The search for a topographic signature of life. *Nature*,
**439**, pp. 411-418.

Fernandez, R., Best, J. and López, F., 2006. Mean flow, turbulence structure, and bed form
superimposition across the ripple-dune transition. *Water Resources Research*, **42**, pp. 1-17.

Fremier, A.K., Yanites, B.J. and Yager, E.M., 2018. Sex that moves mountains: The influence
of spawning fish on river profiles over geologic timescales. *Geomorphology*, **305**, pp.163-172.

Gabbott, S.; Russell, C.; Yohan, Y.; Zalasiewicz, J. 2020. The geography and geology of
plastics: Their environmental distribution and fate. *Plastic Waste Recycling*, pp.33–63.

Gasperi, J., Wright, S.L., Dris, R., Collard, F., Mandin, C., Guerrouache, M., Langlois, V.,
Kelly, F.J. and Tassin, B., 2018. Microplastics in air: are we breathing it in? *Current Opinion*
*in Environmental Science & Health*, **1**, pp.1-5.

GESAMP, 2015. Sources, fate and effects of microplastics in the marine environment: a global
assessment. Kershaw PJ (ed). Rep Stud GESAMP No. 90, pp. 96.

Gottesfeld, A.S., Hassan, M.A., Tunnicliffe, J.F. and Poirier, R.W., 2004. Sediment dispersion
in salmon spawning streams: the influence of floods and salmon redd construction. *JAWRA*
*Journal of the American Water Resources Association*, **40**, pp.1071-1086.

Grant, J., Bathmann, U.V. and Mills, E.L., 1986. The interaction between benthic diatom films
and sediment transport. *Estuarine, Coastal and Shelf Science*, **23**, pp.225-238.

345
346

H., Fang, H.W., Johnson, M.F. and Rice, S.P., 2019. The impact of biological bedforms on near-bed and subsurface flow: A laboratory-evaluated numerical study of flow in the vicinity of pits and mounds. *Journal of Geophysical Research: Earth Surface*, **124**, pp.1939- 1957.

X

Hassan, M.A., Gottesfeld, A.S., Montgomery, D.R., Tunncliffe, J.F., Clarke, G.K., Wynn, G.,
Jones-Cox, H., Poirier, R., MacIsaac, E., Herunter, H. and Macdonald, S.J., 2008. Salmon-

driven bed load transport and bed morphology in mountain streams. *Geophysical Research*
*Letters*, **35**, pp. 1-6.

Hurley, R., Woodward, J. and Rothwell, J.J., 2018. Microplastic contamination of river beds
significantly reduced by catchment-wide flooding. *Nature Geoscience*, **11**, pp.251-257.

Jambeck, J.R., Geyer, R., Wilcox, C., Siegler, T.R., Perryman, M., Andrady, A., Narayan,
R. and Law, K.L., 2015. Plastic waste inputs from land into the ocean. *Science*, **347**,
pp.768-771.

Kane, I.A., Clare, M.A., Miramontes, E., Wogelius, R., Rothwell, J.J., Garreau, P. and Pohl,
F., 2020. Seafloor microplastic hotspots controlled by deep-sea circulation. *Science*, **368**,
pp.1140-1145.

Kennedy, J.F. (1963) The mechanics of dunes and antidunes in erodible-bed channels. *Journal*
*of Fluid Mechanics*, **16**, 521-544.

Lebreton, L., Egger, M. and Slat, B., 2019. A global mass budget for positively buoyant
macroplastic debris in the ocean. *Scientific Reports*, **9**, pp.1-10.

Lee, J., Lee, J.S., Jang, Y.C., Hong, S.Y., Shim, W.J., Song, Y.K., Hong, S.H., Jang, M., Han,
G.M., Kang, D. and Hong, S., 2015. Distribution and size relationships of plastic marine debris
on beaches in South Korea. *Archives of environmental contamination and toxicology*, **69**,
pp.288-298.

Liro, M., Emmerik, T.V., Wyżga, B., Liro, J. and Mikuś, P., 2020. Macroplastic storage and
remobilization in rivers. *Water*, **12**, pp.1-13.

Liu, M., Lu, S., Song, Y., Lei, L., Hu, J., Lv, W., Zhou, W., Cao, C., Shi, H., Yang, X. and He,
D., 2018. Microplastic and mesoplastic pollution in farmland soils in suburbs of Shanghai,
China. *Environmental Pollution*, **242**, pp.855-862.

Ludwig, W., Dumont, E., Meybeck, M. and Heussner, S., 2009. River discharges of water and
nutrients to the Mediterranean and Black Sea: major drivers for ecosystem changes during past
and future decades? *Progress in Oceanography*, **80**, pp.199-217.

McCulloch, M., Fallon, S., Wyndham, T., Hendy, E., Lough, J. and Barnes, D., 2003. Coral
record of increased sediment flux to the inner Great Barrier Reef since European settlement.
*Nature*, **421**, pp.727-730.

Meijer, L.J., van Emmerik et al., 2021. More than 1000 rivers account for 80% of global
riverine plastic emissions into the ocean. *Science Advances*, **7**, pp.1-13.

Morritt, D., Stefanoudis, P.V., Pearce, D., Crimmen, O.A. and Clark, P.F., 2014. Plastic in the
Thames: a river runs through it. *Marine Pollution Bulletin*, **78**, pp.196-200.

Ockelford, A., Cundy, A. and Ebdon, J.E., 2020. Storm response of fluvial sedimentary
microplastics. *Scientific Reports*, **10**, pp.1-10.

Ohata, K., Naruse, H., Yokokawa, M. and Viparelli, E., 2017. New bedform phase diagrams
and discriminant functions for formative conditions of bedforms in open-channel flows.
*Journal of Geophysical Research: Earth Surface*, **122**, pp.2139-2158.

Parsons, D.R., J.L. Best, O. Orfeo, R.J. Hardy, R. Kostaschuk, and S.N. Lane, 2005.
Morphology and flow fields of three-dimensional dunes, Rio Paraná, Argentina: Results from
simultaneous multibeam echo sounding and acoustic Doppler current profiling, *Journal of*
*Geophysical Research*, **110**, F04S03.

Parsons, D.R., Schindler, R.J., Hope, J.A., Malarkey, J., Baas, J.H., Peakall, J., Manning, A.J.,
Ye, L., Simmons, S., Paterson, D.M. and Aspden, R.J., 2016. The role of biophysical cohesion
on subaqueous bed form size. *Geophysical Research Letters*, **43**, pp.1566-1573.

Peng, G., Bellerby, R., Zhang, F., Sun, X. and Li, D., 2020. The ocean's ultimate trashcan:
Hadal trenches as major depositories for plastic pollution. *Water Research*, **168**, pp.1-8.

Pohl, F., Eggenhuisen, J.T., Kane, I.A. and Clare, M.A., 2020. Transport and burial of
microplastics in deep-marine sediments by turbidity currents. *Environmental Science &*
*Technology*, **54**, pp.4180-4189.

Polvi, L.E. and Wohl, E., 2013. Biotic drivers of stream planform: implications for
understanding the past and restoring the future. *BioScience*, **63**, pp.439-452.

Reesink, A.J.H., Parsons, D.R., Ashworth, P.J., Best, J.L., Hardy, R.J., Murphy, B.J.,
McLelland, S.J. and Unsworth, C., 2018. The adaptation of dunes to changes in river flow.
*Earth-Science Reviews*, **185**, pp.1065-1087.

Rezania, S., Park, J., Din, M.F.M., Taib, S.M., Talaiekhosani, A., Yadav, K.K. and Kamyab,
H., 2018. Microplastics pollution in different aquatic environments and biota: A review of
recent studies. *Marine pollution bulletin*, **133**, pp.191-208.

Rice, S., Pledger, A., Toone, J. and Mathers, K., 2019. Zoogeomorphological behaviours in
fish and the potential impact of benthic feeding on bed material mobility in fluvial landscapes.
*Earth Surface Processes and Landforms*, **44**, pp.54-66.

Rice, S.P., Johnson, M.F., Mathers, K., Reeds, J. and Extence, C., 2016. The importance of
biotic entrainment for base flow fluvial sediment transport. *Journal of Geophysical Research:*
*Earth Surface*, **121**, pp.890-906.

Rinaldi, M., Gurnell, A.M., Del Tánago, M.G., Bussettini, M. and Hendriks, D., 2016.
Classification of river morphology and hydrology to support management and restoration.
*Aquatic Sciences*, **78**, pp.17-33.

Statzner, B., 2012. Geomorphological implications of engineering bed sediments by lotic
animals. *Geomorphology*, **157**, pp.49-65.

Syvitski, J., Ángel, J.R., Saito, Y. Overeem, I. .Vörösmarty, C.J., Wang, H. and Olago, D.,
2022. Earth's sediment cycle during the Anthropocene. *Nature Reviews Earth and*
*Environment*, Online First.

Tasseron, P., Zinsmeister, H., Rambonnet, L., Hiemstra, A.F., Siepman, D. and van Emmerik,
422 T., 2020. Plastic hotspot mapping in urban water systems. *Geosciences*, **10**, pp.1-11.

425
426

U th, C.A., Nicholas, A.P., Ashworth, P.J., Best, J.L., Lane, S.N., Parsons, D.R., Sambrook
ns Smith, G.H., Simpson, C.J. and Strick, R.J., 2020. Influence of dunes on channel- scale flow
w and sediment transport in a sand bed braided river. *Journal of Geophysical Research: Earth*
or *Surface*, **125**, pp.1-26.

van Emmerik, T. and Schwarz, A., 2020. Plastic debris in rivers. *Wiley Interdisciplinary*
*Reviews: Water*, **7**, pp.1-24.

van Emmerik, T., Mellink, Y., Hauk, R., Waldschläger, K., Schreyers, L., 2022. Rivers as
Plastic Reservoirs. *Frontiers in Water*, **3**, pp. 1-8.

Venditti, J.G., Church, M. and Bennett, S.J., 2005. On the transition between 2D and 3D dunes.
*Sedimentology*, **52**, pp.1343-1359.

453

. and Xia, B., 2019. Microplastic ingestion in deep- sea fish from the South China Sea. *Science of the Total Environment*, **677**, pp.493-501.

Tables

Table 1 - A table of included plastic particles and their properties

Particle	Material	Density ρ (g/cm ³)	Equivalent Sphere Diameter D (mm) ¹	Submerged Specific Gravity R (-)	Product RD (mm)	Ratio to sand (-)	Observed Mobility
4 mm bead	Polystyrene	1.05	4.0	0.05	0.20	2.89	High
8 mm pony bead	Polystyrene	1.05	8.0	0.05	0.40	1.44	High
8 mm bb gun pellet	Polyethylene	1.25	6.0	0.25	1.50	0.39	High
7 mm Mardis Gras bead	Recycled electrical products	1.24	7.0	0.24	1.68	0.34	Low
9.5 mm Mardis Gras bead	Recycled electrical products	1.24	10	0.24	2.28	0.25	Low
12 mm Mardis Gras bead	Recycled electrical products	1.24	12	0.24	2.88	0.20	Low
14 mm Mardis Gras bead	Recycled electrical products	1.24	14	0.24	3.36	0.17	Low
1 x 1 cm polyester fabric	Polyester	1.38	2.7	0.38	1.02	0.57	High
1 x 1 cm fleece	Polyethylene terephthalate	1.38	2.7	0.38	1.02	0.57	High
1 x 1 cm baby wipe	Polypropylene	1.92	2.7	0.92	2.46	0.23	Low ²
20 x 8 mm cigarette filter tip	Cellulose acetate fibre	1.33	10	0.33	3.31	0.17	Low
0.3 mm PVC powder	PVC	1.36	0.30	0.36	0.11	5.35	High
1 mm thermosetting plastic fragments	Recycled, ground melamine	1.6	1.0	0.6	0.60	0.96	High
0.35 mm sediment	Sand (quartz)	2.65	0.35	1.65	0.58	1.00	High

Notes

¹ Diameter for non spherical particles obtained by computing the particle volume and solving for the diameter of the equivalent sphere

² Behaviour of baby wipes varied due to their properties; they did not maintain their original shape, were torn apart and absorbed fine sediment grains.

Methods

We conducted a 26-hour experiment in a 10m long and 0.5m wide recirculating flume at the University of Hull (Fig. 4). The flume has a maximum depth of 0.5 m and we filled the bottom 0.1 m with sand with a median sediment size D_{50} of 0.35 mm. We filled the flume with water so that the flow depth above the, initially, flat and level sand bed was 0.2 m.

A constant flow discharge of 0.05 m³/s (or 5 L/s) was maintained throughout the experimental sets, giving a depth-averaged flow velocity of the flow of 0.5 m/s. After the flume had been running for 12 hours and the bed had attained equilibrium conditions with the flow, a detailed velocity profile was constructed from measurements over a migrating train of dunes within the central test section of the flume (at 5 m to 7 m from the flume flow entry; Fig. 4) using an Acoustic Doppler Velocimeter (ADV).

After characterizing the flow, a 1.49 kg mixture with 13 different types of plastics, including
 particles, fibres, and fragments, were introduced into the flume. The mass of plastic added to
 the flume only represents 0.12%, by mass, of the total sediment used within the flume system.
 Plastic selected for the study was of different densities and specific gravity (Eq. 1; Table 1) and
 represented commonly used plastic materials. We recognise that the scale of the plastics in
 relation to the size of the morphological unit scale is important in the experimental design. This
 experiment is an end member for small urban streams, which are the most polluting (Meijer et
 al., 2021), and the scaling between the plastic and sand reflects that.

 During the experiments, different particles showed different behaviours in terms of mobility
 under the flow conditions used. The last column in Table 1 shows our observations. To identify
 the threshold between high/low mobility we computed the product of the particle's submerged
 specific gravity (Eq. 1) and its equivalent diameter D and took the ratio with respect to sand.
 The next-to-last column in Table 1 shows how much more mobile the corresponding plastic
 particle is with respect to the sand grains used in the experiment. The threshold between
 high/low mobility for the flow conditions used lies at approximately $RD \sim 1.6$.
 The submerged specific gravity is computed as:

$$R = \frac{\rho_p}{\rho_f} - 1 \quad \text{Eq. 1}$$

Where ρ_p is the density of the particle (see Table 1 for values) and ρ_f is the density of the
 fluid (water in this case).

Equivalent particle diameters for non-spherical particles were computed as:

$$D = \sqrt[3]{\frac{6V_p}{\rho_p}} \quad \text{Eq. 2}$$

482

483

Where V_p is the particle volume. For the fleece, polyester, and baby wipe fragments, a thickness of 0.1 mm was assumed.

Figure 4 in

interactions. This included video capture (see supplementary material) and detailed mapping
 of the transport pathways.

 *Figure 4 – (A) Side-view and (B) cross-section view of recirculating flume; (C) schematic of*
 *sand and plastic particles used in the experiments, and (D) flow velocity measurements and*
 *profile.*

*Table 2 - A table of included plastic and their densities*

Particle	Material	Density ρ (g/cm ³)	Equivalent Sphere Diameter D (mm) ¹	Submerged Specific Gravity R (-)	Product RD (mm)	Ratio to sand (-)	Observed Mobility
4 mm bead	Polystyrene	1.05	4.0	0.05	0.20	2.89	High
8 mm pony bead	Polystyrene	1.05	8.0	0.05	0.40	1.44	High
6 mm bb gun pellet	Poly lactide	1.25	6.0	0.25	1.50	0.39	High
7 mm Mardis Gras bead	Recycled electrical products	1.24	7.0	0.24	1.68	0.34	Low
9.5 mm Mardis Gras bead	Recycled electrical products	1.24	10	0.24	2.28	0.25	Low
12 mm Mardis Gras bead	Recycled electrical products	1.24	12	0.24	2.88	0.20	Low
14 mm Mardis Gras bead	Recycled electrical products	1.24	14	0.24	3.36	0.17	Low
1 x 1 cm polyester fabric	Polyester	1.38	2.7	0.38	1.02	0.57	High
1 x 1 cm fleece	Polyethylene terephthalate	1.38	2.7	0.38	1.02	0.57	High
1 x 1 cm baby wipe	Polypropylene	1.92	2.7	0.92	2.46	0.23	Low ²
20 x 8 mm cigarette filter tip	Cellulose acetate fibre	1.33	10	0.33	3.31	0.17	Low
0.3 mm PVC powder	PVC	1.36	0.30	0.36	0.11	5.35	High
1 mm thermosetting plastic fragments	Recycled, ground melamine	1.6	1.0	0.6	0.60	0.96	High
0.35 mm sediment	Sand (quartz)	2.65	0.35	1.65	0.58	1.00	High

Notes

¹ Diameter for non spherical particles obtained by computing the particle volume and solving for the diameter of the equivalent sphere

² Behaviour of baby wipes varied due to their properties; they did not maintain their original shape, were torn apart and absorbed fine sediment grains.

**Acknowledgements**

The authors are thankful for the help offered by Connor Burchell, Marijke de Vet, Stuart
McLelland, Greg Smith, and Clint Wilson during different stages of our experiments, including
setup, measurements, and clean up. Participation of RF was made possible thanks to funding
by the European Research Council under the European Union's Horizon 2020 research and
innovation program (grant 725955), and the Leverhulme Trust, Leverhulme Early Career
Researcher Fellowship (grant ECF-2020-679).

**Author Contribution Statement**

C.R. conceived and led the project. C.R., R.F. and S.E.G. designed the experiments and
methodology. C.R. and R.F. performed the experiments and analysed the data with
contributions by D.P. C.R. wrote the first manuscript draft and contributed Figures 1-3 and the
Supplementary Materials, and RF wrote the Methods and contributed Figure 4. All authors
discussed results and implications and commented on the manuscript at all stages.

**Competing Interest Declaration**

The authors declare no competing financial interests.

**Supplementary Material**

Supplementary Information is available for this paper.

*Description of supplementary footage is provided below. The supplementary footage is*
*provided as an .MPG file titled "Russell et al., Supplementary Footage".*

**Clip 1** (00:00:00-00:00:08)

Flow direction is from right to left

**Description:** Migration of uncontaminated dunes. Sand is eroded from the stoss side and
redeposited on the lee side of the dune, such that it steadily migrates downstream.

**Associated Figures:** Fig. 1A

**Clip 2** (00:00:08-00:00:27)

Flow direction is from left to right

**Description:** 12 mm Mardis Gras bead sits on the lee slope of the migrating dune in incipient
motion until it is stabilized and covered by sand and buried as an individual particle.

**Associated Figures:** Figs 1Bii; 1Biii

**Clip 3** (00:00:27-00:00:37)

Flow direction is from right to left

**Description:** 14 mm Mardis Gras bead at the bottom of a lee slope rolls around in the
recirculation zone prior to incorporation and burial at the base of the lee slope.

**Associated Figures:** Fig. 1Bii

**Clip 4** (00:00:37-00:00:59)

Flow direction is from right to left

**Description:** A group of low mobility plastic particles cluster, deflecting the flow and
allowing higher mobility particles to aggregate and become buried.

**Associated Figures:** Fig. 1Biii

**Clip 5** (00:00:59-00:01:16)

Flow direction is from right to left

**Description:** Plastic particles with high mobility accumulate to a sufficient level to undertake
multi-layered deposition.

**Associated Figures:** Figs 1Bi; **2A**

**Clip 6** (00:01:16-00:05:29)

Flow direction is from right to left

**Description:** A recently initiated dune begins to accumulate plastic on its lee slope, which
then becomes incorporated into the dune as it advances. Notably, two cigarette filters are
incorporated and later eroded (at 00:03:57 and 00:04:42), causing the dune to adjust, reduce
its volume, and wash out. The pit formations induce the formation of small bedforms that
migrate up the stoss slope to the crest. Note at 00:3:42, a string is formed by an 8 mm bead
that has a 4 mm bead stuck in it; the combined properties of the beads caused it to behave
differently with (a higher RD value) and promote the deposition of high mobility plastic
particles.

**Associated Figures:** Figs 1B; 1C; 1D; **2C**

**Clip 7** (00:05:29-00:06:34)

Flow direction is from right to left

**Description:** *Clips 7 and 8 are from the same dune.* A group of plastic particles is eroded
from the stoss side of the dune causing an increased rate of erosion and slope steepening.

**Associated Figures:** Figs 1Ci; 1D; **2B**

**Clip 8** (00:06:34-00:07:49)

Flow direction is from right to left

**Description:** *Clips 7 and 8 are from the same dune.* From the beginning of the clip, the slope
steepening caused by erosion of the group of plastic particles in Clip 7, is seen to migrate
towards the dune crest. The resulting dune shape is significantly more symmetrical and lower
in volume.

**Associated Figures:** Figs 1Ci; 1D; **2B**

**Clip 9** (00:07:49-00:08:28)

Flow direction is from right to left

**Description:** Changes to dunes upstream secondarily impacts this dune as it adjusts its shape
to re-equilibrate to the flow conditions via a symmetrical dune form.

**Associated Figures:** Fig. 1D

**Clip 10** (00:08:28-00:10:00)

Flow direction is from right to left

**Description:** Plastic particles become exposed on the stoss side of a migrating dune and
erosion is enhanced around the obstacle. Whilst the particles themselves are not eroded out of
the bed, the comparative scales of the obstacle and dune see that the dune becomes washed
out.

**Associated Figures:** Figs 1Ci; 1D

Reviewer # 1 Comments (and author replies):

The presence of plastic pollution in river ecosystems dramatically increased in the last decades. The paper deals with an emergent topic which gained lots of interest from the scientific community, including a very wide field of applications and base research.

In particular, we recognized that plastic particles undergo transport process in some way similar to sediment particles, but with several important differences which attain to shape, density and roundness of plastic particles.

The experiment is sufficiently described, the video provided in the Supplementary Materiale are very nice and help to understand the processes. Some details of the experiment need to be better described. After my review of the paper, my recommendation is "Accept with Minor revision".

Dear Reviewer:

Thank you very much for your time to review and for your positive assessment of our manuscript. We have incorporated your suggestions and addressed the comments you have made. Thanks for helping us improve the clarity and reproducibility of our work!

Some suggestions follow.

1. The introduction is missing on the discussion of previous works regarding transport processes experienced by plastic particles. The introduction will benefit from the addition of some of the recent works on plastic transport, which investigated settling velocity (see among the others Khatmullina and Isachenko (2017), Waldschlager and Schüttrumpf (2019a), Francalanci et al. (2021)), the influence of shape and density in transport processes, and the influence of the threshold of incipient motion (see for example Waldschlager and Schüttrumpf, 2019b), but there are several others references that can be added.

We have added the references suggested in the 'Introduction', with corresponding text to cover this background. Although we do not go into specific details, we recognize the recent contributions of the community.

2. Method section. I do not understand why the "method" section is after the references. At the beginning I thought it was an intentional choice, but after my review, and given the short length of this section, I believe it was a mistake with the document. I recommend moving the method section after the introduction section. The information reported in the Method section are essential to the reader for understanding the results and the discussion, they are not optional, so the paper needs to be re-organized.

Due to the style of this journal, methods are included after the conclusions. We have moved them to be before the references and included a summary of the methods in the 'Introduction'.

3. Results and Discussion. Lines 101-112. Some details of the experiments are anticipated here, but this short description is not sufficient for the reader to have a minimum understanding of what the experiments is about: how much is the percentual of plastic in the sand? How many experiments did you run? Which are the characteristics of plastic particles used in the experiment? The reader is forced to skip the text and look at the end of the paper for this information, in order to the results and discussion section.

These additional information should be added to the experiment description.

Moreover, Table 1 is cited here (Line 106, 112) but is much farther in the text. My suggestion is a better structure of the paper, reorganizing the sections, Figures and Table.

We have extended the 'Methods' section and placed it in accordance with the journal style. Additions include additional relevant parameters and challenges with scaling and ground truthing.

We have also added a few sentences within the 'Introduction' to describe our materials and methods and point the reader to the corresponding section for further details.

"Therefore, we designed and undertook physical laboratory experiments in a 10 m long, 0.5 m wide recirculating flume (Figure 4A). We added enough sand (1,241.6 kg) with a median size (D_{50}) of 0.23 mm (Figure 4D) to create a 0.1 m thick deposit over the entire length of the flume. The flume was filled with water to a depth of 0.2 m above the sand bed and turned on to convey flow at a mean velocity of 0.5 m/s (Figure 4C). After the sand bed had developed dunes that were in equilibrium with the flow (12 hours) we added a mixture of plastic materials (1.49 kg) of different densities and sizes (Figure 4D and Table 1) and documented the interactions between the plastic particles and the sand particles over the channel bed for more than 12 hours. These experiments were designed to explore whether plastic influences riverbed sand transport mechanisms and focused on representing small urban stream systems, which are commonly highly contaminated with plastics of different types due to their proximity to pollution sources (Meijer et al., 2021). A full description of the materials and setup is available in the Methods section."

Thank you for helping us address this lack of clarity and details within the manuscript.

4. The presentation of the results is very clear and informative. I suggest citing the Video Clip number when discussing the observed processes. The Supplementary Material is very important to understand the processes described in the paper.

We have added these references as recommended.

5. Lines 454 and following. How is the plastic mixture composed? The total weight is not enough to reproduce the experiment. The authors correctly said that "the scale of the plastics in relation to the size of the morphological unit scale is important in the experimental design", so how do you scale the size, density and quantity of plastic observed in the urban stream in Meijer et al., 2021?

We have added all the details to Table 1.

As it is a complex challenge to directly observe the bottom of a river, the only information presently available is from studies looking at plastic concentrations in the water column. For example, Morritt et al., 2014 used nets about 40cm from the bottom of the River Thames (<https://doi.org/10.1016/j.marpolbul.2013.10.035> - last accessed 16-Sep-2022), but we do not know how much of this was interacting with the riverbed, and to what extent. Most studies on riverine plastic pollution are of floating plastic (e.g., various works of Tim van Emmerik), so material that is actually affecting the riverbed is unknown. Meijer et al. (2021), specifically discusses that small urban streams are the most polluted, and this is why we cite that paper.

Plastic particles were selected to incorporate a range of densities and sizes frequently encountered in littered plastics so that a range of mobility values (quantified through the product of the submerged specific gravity and particle diameter RD) could be assessed. The selection of plastic particles are a starting point from which we will continue to refine in our next experiments, in which we will further define the relative impact of different types of plastic particle on the bedforms.

We have summarized this and added it to the text:

“Directly observing sediment dynamics within a natural river bed is a significant challenge, meaning that ground truthing experimental results in the field is complex. A range of approaches have been attempted such as Morrill et al. (2014) where nets were anchored to the bottom of a river. However it was only possible to position these nets ~40 cm above the river bed, meaning the interactions of plastic with the bed remained uncertain and largely unquantified. Herein we used a range of generic plastic types found in river systems, including both elongate (cigarette filter tips) and rounded shapes of different sizes and densities (Eq. 1; Table 1). Rounded shapes readily represent the endemic challenge of nurdle spills (de Vos et al., 2021), which although they are most studied in marine settings, also occur along rivers, e.g., the Mississippi River. Additionally, rounded shapes are an ideal and readily relatable starting point for translating existing knowledge of sedimentary mechanics to find the knowledge gaps for further study.”

6. Moreover, if I understood correctly the authors carried out only one experiment: I am not sure that with only one experiment they can address general conclusion; they can describe what they observed but a relevant number of questions remain unresolved: which is the influence of the composition of the plastic mixture added to the sand? Do you try to repeat the experiment to verify the results? Can you expect similar results if you change plastic percentual and plastic composition?

We fully understand the reviewer’s concern and acknowledge that we only used one set of flow conditions. However, it might be deceiving to suggest that we only did one experiment as we conducted observations of different dunes within the flume for more than 12 hours (dozens of dunes and 1000’s of observations). Such controlled constant flow and transport conditions in the field would likely be impossible to obtain for such a long period. The experiment ran for 26 hours: 12 hours to guarantee that the clean sand bed was in equilibrium with the flow conditions; 2 hours to conduct measurements for flow characterization and to add the plastics; and over 12 additional hours of observations.

It is true that we only used one flow condition, but during our observations, we saw the same lee-side processes occurring at different locations (many different bedforms) and repeatedly, therefore these conclusions are from dozens of observations. We have added the weight of each plastic type to Table 1 and a note in the text stating that some plastic particles were more influential on the bedload than others (such particles are also indicated with a star * in Table 1). Lastly, the comment on changing plastic percentual and plastic composition causing different changes is an inference that extends beyond our present dataset and would require further study. We have added a note in the text to this effect.

Minor points.

Figure 4. Move the Arrow “Flow” on the left to a better spot; now it seems connected in some way with the tilting mechanism. It would be better to move it below the recirculating pipe.

We have moved the flume arrow to underneath the ‘Return pipe’ label for clarity.

Lines 451-452 – Can you add the location of the test section in the flume in panel a) Figure 4?

We have added a label to indicate the reach where we conducted the observations.

Lines 463-467. It took me a while to understand that RD is R multiplied by D (diameter). You can rephrase the sentence, for example: “to identify the threshold We computed the parameter RD as the product of R, particle’s submerged , and it equivalent diameter D ...”

In the table change "Ratio to sand (-)" with "RD ratio to sand (-)"

This has been amended.

Table 1 and Table 2 are the same.

This has been amended. There should be no Table 2.

References

Khatmullina, L., Isachenko, I., 2017. Settling velocity of microplastic particles of regular shapes. Mar. Pollut. Bull. 114, 871–880. <https://doi.org/10.1016/j.marpolbul.2016.11.024>.

Waldschlager, K., Schüttrumpf, H., 2019. Effects of particle properties on the settling and rise velocities of microplastics in freshwater under laboratory conditions. Environ. Sci. Technol. (53). <https://doi.org/10.1021/acs.est.8b06794>, 1958–1966.

Waldschlager, K., Schüttrumpf, H., 2019. Erosion Behavior of Different Microplastic Particles in Comparison to Natural Sediments. Environ. Sci. Technol. 2019, 53, 13219–13227. <https://doi.org/10.1021/acs.est.9b05394>.

Francalanci, S., Paris, E., Solari, L. (2021). On the prediction of settling velocity for plastic particles of different shapes. Environmental Pollution, 2021, 290, 118068. <https://doi.org/10.1016/j.envpol.2021.118068>

Thanks for sharing these!

Reviewer # 2 Comments (and author replies):

Comments by J.H. (Janrik) van den Berg

28-5-2022

Dear Dr. van den Berg:

Thank you very much for the time you spent reviewing this manuscript and for sharing your thoughts with us. We greatly appreciate your feedback and have worked on the manuscript, following your recommendations and believe that it is now clearer and more tempered in its implications. This revised version presents the results of our small-scale laboratory experiments and while it addresses your comments about local effects and scaling concerns, it relates them to natural environments in the context of morphodynamic-similarity (Paola et al., 2009). We hope that you now agree that its contributions to the field are better constrained while presenting the challenges and opportunities appropriately. We look forward to your thoughts on this revised version. In the following paragraphs, we present detailed answers to all your comments and queries.

This might become a valuable contribution as it contains novel information on the influence of plastic particles on dune dynamics. However, its far-reaching claims of its value to nature are not justified by the experiments executed, because they do not reflect conditions found in many natural rivers. I would therefore advise the authors to restrict their conclusions to the results of the experiments and to be humble and very reserved about their generic value.

Throughout, we have made it clearer that we are only observing local environments, and the conclusions have been modified where the authors agreed that it was necessary, e.g., the headings of the key findings have been modified to include the caveat that the observations are local only, so that we are not overstating our findings.

"2) Plastic inclusion locally changes the ratio of suspended load to bedload material"

"3) Inclusion of plastic encourages rivers to locally develop more conduit-like properties and create heterogeneous deposits."

We have only mentioned the rate of transport where the previous steps of local increase of sand into suspension is stipulated and we explain this more clearly now, as examined in a comment below.

Additionally, two additional paragraphs have been added to Key Findings that outline the limitations of the study with regards to ground truthing our particle selection, and scaling.

A lot of the text, especially in the Introduction, is not essential to what should be the main objective of the paper: description and analysis of "the effect of a mixture of plastic particles on bedform dynamics in some flume experiments" ("..." suitable title of the paper) and therefore can – and should - be omitted.

We fully agree that a large proportion of the Introduction has information that would be very basic for scientists familiar with the topic. However, we recognize the broad reach of Communications Earth and Environment and therefore have opted to keep in the sedimentological background information to widen the reach of our paper and to ensure that non-experts are also capable of understanding our work and its implications.

Regarding the title, our title originally was "Plastic pollution in riverbeds fundamentally affects natural sand transport". The options that the referee suggest are too technical such as including "bedform dynamics" and "flume tank"; will not be understood by a broad audience and so we have not done this.

This referee makes an additional in-text concern, stating that the plastic affects the transport is not to imply a rate. We do not mention that it increases the transport rate, just that it affects natural sand transport.

In the manuscript we clearly demonstrate that plastic inclusion affects the mechanics of the downstream migration of dunes, whereby locally more sand is entrained into suspension rather than being in bedload. As such, the physical transport processes by which sand travels are affected and we have added “processes” to the end of the title to clarify that we are not inferring a rate of change, simply the mechanics. The title now reads: “Plastic pollution in riverbeds fundamentally affects natural sand transport processes”.

Conversely, text and figures should be added to (1) document better the experiments

This is a fair point. We have included a graph of grainsize distribution to Figure 4 (see Figure 4E), added the amount of each plastic particle to Table 1, and referenced the supplementary videos in the text. We have also included relevant parameters in the ‘Introduction’ and ‘Methods’ sections for the interested reader (e.g. Fr and Re numbers as well as bed shear stresses and Shields number).

(2) integrate information on the effect of natural low-density admixtures (organic debris) on dune dynamics.

Organic debris and plastic have low density in common, though they are not analogous. We have added information about previous studies on plastic in water as recommended by Reviewer 1, and included mention of studies of organic material at several places through the manuscript. It would be a separate study to assess the similarities and differences of organic material and plastic, and is beyond the scope of this manuscript.

We understand that the goal here is to guide us to potential ground truthing analogues, however, the novelty of this experiment means that there is very little data to truly draw on within these contexts. There are many further clarifications, tests, and further studies that can be done as plastics are a new frontier in sedimentology, and this study has got important implications for its advancement.

(3) discuss to what extent the experiments represent natural conditions and to adapt the conclusions accordingly.

We think that this comment may stem from this reviewers concerns over the applicability of laboratory experiments to real-world river systems – a subject we have expanded on in detail above and in the manuscript with reference to Paola et al. (2009, <https://doi.org/10.1016/j.earscirev.2009.05.003>) where appropriate.

If we were to replicate a real river for the first phase of the experiments, there would be too many variable to understand the impacts of plastic in the system. Therefore, this experiment is to specifically test what effect plastic has on bedforms under simple flow rates and conditions (i.e. without meanders and depth changes), so that we can begin our understanding of how the inclusion of plastic items changes the way in which sand is transported on a river bed.

As such, we have not intended to replicate a river, we have replicated a riverbed in a simple and controlled channel with plastic particles included.

However, we take on board the reviewers concerns about over-extending our conclusions. Thus, we have retained our interpretations and highlight that our analyses are valid under the context of morphodynamic-similarity. We mention the limitations and we have also provided explanation regarding the limitations of applying lab data to real systems. This is all explained in the detailed answer above regarding this issue.

In a flume with a water depth of two dm only small dunes can develop, much smaller than usually found in a river. In fact, the bedforms studied are only up to about 5 cm high, which is only slightly higher than the size of ripples and smaller than any natural dune found developed in conditions where size is not limited by water depth. With larger, more natural dunes, accumulations of plastic in the dune trough and its effect on the dune shape may have a much smaller effect as observed in the experiments, because those dune troughs are much larger and deeper.

We agree with the reviewer that the scale of the plastics and the scale of the dunes would not reflect all scenarios found in natural environments. Nevertheless, it does provide new important data on what happens at the scale we tested. To take on board the reviewers comment, we have added this limitation to the manuscript and let the reader know the scaling challenges, but that this fundamental disruption to grain-to-grain interactions will remain valid in the context of larger dunes, and that this still needs to be studied further.

An indication of this is the influence of organic detritus on crossbedding as recorded in some previous literature. I would recommend going into this and pointing out the similarities/differences with the interaction by plastic debris. Some examples can be found in the book I wrote with Allard Martinius "Atlas of sedimentary structures in estuarine and tidally-influenced river deposits of the Holocene Rhine-Meuse-Scheldt system" (e.g., page 107, figure 3.6.1; page 128, figures 4.1.9 and 4.1.10). Organic material is deposited in the toe of the dunes during periods of low flow speed (see also Martinius and Gowland, 2010: <https://doi.org/10.1111/j.1365-3091.2010.01185.x>). This raises the question under which experimental conditions of dune formation the accumulations of the plastic particles occurred. In other words, where do the experiments plot in stability diagrams, just above the start of dune formation?

This is an interesting point and could perhaps be an area of fruitful further research – however, addressing this is beyond the scope of the experiments we designed and conducted. However, we will make the following comments and have added some detail to take on board this point on low density organic material. Prior studies on organic debris that is entrained into dunes in tidal settings (e.g. Martinius and Gowland, 2010; Martinius and Van den Berg, 2011), rely on the mechanism that the changes in flow speed encourage the deposition of the organic material. The organic material picks out spring and neap cycles, which themselves represent variations in waxing and waning flow, thereby they are not representative of what has occurred in this study. We do acknowledge this aspect and reference the work you have shared with us in the 'Results' section. This study represents a unidirectional, entirely non-tidal river system that exists at a constant flow throughout the entire experiment time. However, we see the value of inclusion of organic material as a comparison, due to its lower density and therefore potential similarities with plastic transport. A paper on the inclusion of waterlogged charcoal has been integrated into our discussion and we address the issue in the 'Introduction' (Nichols et al., 2000 - Nichols, G.J., Cripps, J.A., Collinson, M.E. and Scott, A.C., 2000. Experiments in waterlogging and sedimentology of charcoal: results and implications. *Palaeogeography, Palaeoclimatology, Palaeoecology*, 164(1-4), pp.43-56.). This paper undertook experiments in unidirectional constant flow, allowing for an organic comparison.

We have included the plot (phase-diagram) in an answer to a comment below (see answer to your comment to line 102 of the original submission).

Another concern in these experiments is to what extent the material choice and concentration approach reality. Is the mixture representative of conditions in nature in terms of particle shape and grain size distribution? To be able to test this in future research, it is necessary that the authors give grain distributions of the mixtures used.

[As outlined above for Reviewer 1]

As it is a complex challenge to directly observe the bottom of a river, the only information presently available is from studies looking at plastic concentrations in the water column. For example, Morrill et al., 2014 used nets about 40cm from the bottom of the River Thames (<https://doi.org/10.1016/j.marpolbul.2013.10.035> - last accessed 16-Sep-2022), but we do not know how much of this was interacting with the riverbed, and to what extent. Most studies on riverine plastic pollution are of floating plastic (e.g., various works of Tim van Emmerik), so material that is actually affecting the riverbed is unknown. Meijer et al. (2021), specifically discusses that small urban streams are the most polluted, and this is why we cite that paper.

Plastic particles were selected to incorporate a range of densities and sizes frequently encountered in littered plastics so that a range of mobility values (quantified through the product of the submerged specific gravity and particle diameter RD) could be assessed. The selection of plastic particles are a starting point from which we will continue to refine in our next experiments, in which we will further define the relative impact of different types of plastic particle on the bedforms.

We have summarized this and added it to the text:

“Directly observing sediment dynamics within a natural river bed is a significant challenge, meaning that ground truthing experimental results in the field is complex. A range of approaches have been attempted such as Morrill et al. (2014) where nets were anchored to the bottom of a river. However it was only possible to position these nets ~40 cm above the river bed, meaning the interactions of plastic with the bed remained uncertain and largely unquantified. Herein we used a range of generic plastic types found in river systems, including both elongate (cigarette filter tips) and rounded shapes of different sizes and densities (Eq. 1; Table 1). Rounded shapes readily represent the endemic challenge of nurdle spills (de Vos et al., 2021), which although they are most studied in marine settings, also occur along rivers, e.g., the Mississippi River. Additionally, rounded shapes are an ideal and readily relatable starting point for translating existing knowledge of sedimentary mechanics to find the knowledge gaps for further study.”

Additionally, we have added the proportions of plastic particles to Table 1

It is also necessary to provide a useful measure for the concentration, 0.12% by mass of the sand present in the flume is not. I propose the volume percentage in relation to the sand in the active layer (= dune volume) per meter of stream length and width.

We presented the concentration in a way that is meaningful to researchers sampling for microplastics in the field and to allow others to replicate our experimental setup. There are many different ways of quantifying the abundance of plastics and as yet a consensus unit has not been established. We note that many plastics researchers will be unfamiliar with the concept of active layer.

Moreover, out of the initial content of plastics added, not all of them interacted with the active layer and we did not specifically quantify the mass of plastics interacting with the active layer, or the amount of plastics trapped underneath. We observed and documented the types predominantly interacting with the active layer (see Table 1) but it would not be fair to say that all mass of such plastics added was affecting the bed transport mechanics and presenting such concentration.

Additional comments I have inserted in the margin of the MS transferred to Word.

Thanks for your detailed comments within the manuscript. We have copied those comments in here and referenced the line numbers in the original submission for clarity.

Line 1: Title incorrect, because sand transport was not measured! Suggestion for title that better fits the subject: "Influence of plastic particles on dune dynamics in some flume experiments"

We agree that the transport was not measured, but we prefer to continue to use a title that is meaningful for a wide audience, including non-experts. As such, we propose to keep our original title because it is simple language without technical or subject-specific language. Additionally, from the findings, it does impact the bedload processes, thereby affecting the transport, whether or not the measurements were taken. We have added "processes" to the end of the title to clarify that we are discussing changing mechanics, not rates.

Lines 9-10: Omit, because not essential to the subject

We agree. Omitted on the reviewers recommendation.

Lines 19-22: This far-reaching conclusion cannot be drawn from the experiments!

We have rephrased the sentence to temper the conclusion and encourage the reviewer to consider the widely accepted concept of morphodynamic similarity summarized in great form by the authors in Paola et al., 2009: *The "unreasonable effectiveness" of stratigraphic and geomorphic experiments*. Link: <https://doi.org/10.1016/j.earscirev.2009.05.003>. Small-scale experiments are suitable to infer natural processes when they involve sediment transport processes as the underlying physics express themselves despite the changes in spatial and temporal scales.

Lines 49-55: You cannot extent conclusions from experimental results to nature if you don't know whether experimental conditions approach nature.

We prefer to trust in the well-documented experience of so many researchers who have been able to study natural processes and better describe them by using small-scale experiments. As in the comment above, we encourage the reviewer to check Paola et al., (2009).

Lines 57-62: Well-known to everybody, can be omitted

We agree that this is well known to everyone in the sedimentology community, however, this manuscript is intended for a journal that reaches a far broader readership, so we need this to help the audience to follow along.

Lines 65-72: Not really relevant to this paper, can be omitted

We have made the connection to the narrative clearer in the text.

The authors feel that highlighting how something such as biotic effects can have seemingly small-scale impacts on rivers that escalate to large-scale landscape changes is very important when looking to discuss the potential changes that a small change in the process sedimentology can have on the landscape. One great example (that we cite) is associated with the spawning activities of salmon and how this affects long-term river profile shapes (See: Fremier et al., (2018): Sex that moves mountains: The influence of spawning fish on river profiles over geologic timescales). We cite additional relevant work.

Lines 78-88: This is the basic information needed. In fact all other information given in this Introduction is not adding much to this and therefore in my opinion can be left out.

Again, we would agree with you if this were a journal specific to sedimentologists. However, we consider the readership of Communications Earth and Environment to be broad and therefore this is necessary context for the average reader that we expect this manuscript to encounter.

Lines 91-96: This is basic info, known by everybody, can be left out

We prefer to retain this information so that the manuscript can be understood by the broad readership of Communications Earth and Environment.

Line 97: ...the well-established flow and sediment transport processes over dunes...

Thank you, this has been amended.

Line 101: D50 and D90 of 0.35 mm and ... mm resp., see also Fig. ... (=grainsize distribution = NEW FIG.)

We agree, we have kept the text similar but added a graph of the grain distribution to Figure 4 and referenced it here.

Line 101: Provide graph of grainsize distribution

Please see new Figure 4 (specifically 4D)

Line 102: This is not a useful parameter. Provide information such as average values of flow, bed shear stress, Shields parameter etc and show how the experiments plot in well-known stability diagrams (do not forget to correct values for flume side-wall roughness).

We now include relevant dimensionless numbers (Fr , Re , Shields) in the text ('Introduction' and 'Methods') and have also added the shear stress values. We have corrected the shear stresses using the approach of Vanoni and Brooks (1957) with the code developed by Fernández et al. (2021). Using the hydraulic radius associated with the bed region only, to compute the Shields number, and the corresponding particle Reynolds number, we fall slightly below the region for natural sand-bed streams in the Parker Shields diagram. Our bedforms are borderline between ripples and dunes. We have chosen to continue to refer to them as dunes as the readership of Earth Communications and Environment is expected to be more familiar with the term 'dune' than with the alternative terms 'ripples', or 'bedforms'. Also, inclusion of the phase diagram only to show where our experiments fall seems unnecessary in the manuscript. We include it below for the reviewer's reference:

(Screenshot from Figure 2-28 of the Engineering Sedimentation Manual edited by M.H. Garcia (2009) with our results marked by blue lines)

A 1°C variation in water temperature would change the viscosity enough to shift our results slightly. This is also true for any change in the submerged specific gravity of the sand used. We assumed $R = 1.65$ as is commonly done but there is a range and it could also

shift our result. Given that is borderline and that the term 'dune' is better recognized by non-specialists, we have chosen to continue to use it.

Line 115: ... from, very small experimental dunes

We have included "small" to help the reader with further interpretations of the text/implications of our observations. The scaling is also available in the figure itself to further indicate the scale of the experiments.

Lines 204-206: A reduction in dune volume results in a higher migration rate, not necessarily in a higher sediment transport.

If the sand is migrating downstream more rapidly, it is being transported more rapidly, and even if this is only locally, it will be reducing the overall transport time of the sand. In the videos that accompany the manuscript, these bursts of sediment into suspension can be readily observed. Whilst we do not have enough information specifically from this experiment here to say that one dune burst will change the entire system, we can and are showing that the plastic is encouraging more sand to change from near bed to suspended, which can infer from the impact of biota and Paola et al., 2009, that the changes will scale up. We know that more sand is in suspension because of the plastic, so particles are moving further in less time, and whilst this is not yet quantified, we can see clearly from the videos that it is indeed happening and the effect is logical as outlined in this sentence. We have rephrased for clarity:

"In all cases, the pits and scours migrating towards the dune crest reduce the dune volume, increase the downstream migration rate, and suggest that a local increase in the overall rate of downstream sand transport is likely (Figs 1Cii; 1Dii)."

Lines 206-209: If true in nature this would have two important consequences you might address:

1. With the hydraulic roughness would become much lower, resulting in less deep rivers (with possible consequence for navigation at low discharge and (less) flooding at peak discharge. Before making this conclusion generic, please check have such changes been recorded in rivers with plastic pollution?

2. when the separation vortex disappears, the hydraulic form roughness also disappears, leaving more energy for sand transport.

Thanks for your very useful observations that have further contributed to the potential implications of our study. We have incorporated these comments into this paragraph and into the key findings near the end of the manuscript. This now reads:

"This is critically important because it demonstrates for the first time that plastic can fundamentally change the local conditions and sediment transport mechanics at the riverbed, which may in turn alter its suitability for biotic habitation and impact longer-term river channel evolution, i.e., the observed flattening of riverbed topography lowers the hydraulic roughness (resistance to the flow offered by the riverbed), which could over time result in shallower rivers. On the other hand, if the disruptions to the bed resulted in shorter but taller bedforms, the hydraulic roughness would be higher and flood stage would increase. This aspect needs further investigation."

And:

"As the crest of the dune becomes more rounded or flattened, the angle of the lee slope may become too shallow to sufficiently shelter sediment for deposition in the lee side, or it may become entirely indistinct, such that the separation vortex in the recirculation zone disappears and the dune is washed out (Fig. 1Dii)."

Your thoughts outline two very much needed avenues of research within this topic and we have plans to pursue them in the near future.

Line 253: Restrict your findings to your experimental results as you have no any "ground truth" with natural rivers

We accept this to be a limitation of the work and state this explicitly within the manuscript. We write:

"Observing riverbed dynamics in the field is very difficult, particularly if the observations must be carried out without disturbing the occurring processes. To overcome this, fluvial geomorphologists have used small-scale experiments in the laboratory, for many decades, and it is widely accepted that these kind of experiments have an 'unreasonable effectiveness' (Paola et al., 2009) due to natural scale independence, i.e. the underlying physics express themselves despite the difference in the spatio-temporal scales. Our experiments, like many others, are imperfect and limited, but the observations are likely consistent with field systems thanks to the well-documented and accepted morphodynamic similarity principle (e.g. Paola et al., 2009). Additionally, natural river systems are inherently complex, therefore flume tank experiments offer an excellent opportunity to isolate a variable, such that we can better understand that component of the complexity."

However, this study was required as the first step to understand what we need to look for in natural systems, therefore it is beyond the scope of this study to do ground truthing. It is reasonable to expect these processes to exist in nature [see Paola et al., 2019 <https://doi.org/10.1016/j.earscirev.2009.05.003>].

Lines 261-263: The experiments only show a reduction of dune volume, not a change in sediment transport

We have edited this section to remove the implication of rate and transport, but we maintain that the process of plastic being eroded from the stoss side of the bedform, changes the mechanics of the dune, thereby reducing the dune volume.

"The speed of bedform morphological transformation was found to increase with the addition of plastic, which affects bed topography and enhances dune erosion"

Additionally, this is where we have added your comment on lower riverbed roughness potentially resulting in shallower rivers:

"This is critically important because it demonstrates for the first time that plastic can fundamentally change the local conditions and sediment transport mechanics at the riverbed, which may in turn alter its suitability for biotic habitation and impact longer-term river channel evolution, i.e., the observed flattening of riverbed topography lowers the hydraulic roughness (resistance to the flow offered by the riverbed), which could over time result in shallower rivers. On the other hand, if the disruptions to the bed resulted in shorter but taller bedforms, the hydraulic roughness would be higher and flood stage would increase. This aspect needs further investigation."

Line 267: You have not done any measurement of bedload transport!
Lines 268-269: You have not made any measurement of suspended bed-material transport!

Whilst we did not measure the bedload transport, we did find and film these changes locally, such that the statement "plastic inclusion changes the ratio of suspended load to bedload material" is indeed correct.

Lines 267-273: The introduction of plastic particles in a flow might damp turbulence which would counteract the increase of sand transport you suggest. So, in absence of any support from measurements your claim of increased sand transport is speculation.

We agree with the reviewer that there is extensive work on turbulence dampening by the presence of suspended sediment. We also agree that we cannot say there is an increase in sand transport at the reach scale. The amount of sand particles clearly switching from bedload transport to suspended sediment transport due to the presence of plastics (see Video Clips for details) might or might not be enough to dampen turbulence. The videos do show localized events where plastics lead to the destruction of dunes and their entrainment into the flow leads to increased amounts of sand in suspension. We did not quantify this, but the videos are clear in what they show and one can derive inferences from them, or as you say, speculate.

Lines 275-276: This may be true, but not based on your experiments, and therefore should not be part of this chapter

We have added the caveat of "locally" to the sub-title, such that the implications are more constrained. We agree that rivers change their properties downstream, therefore we are focused on the local changes that we saw.

Lines 286-291: May be correct, but not based on data and rather far beyond your experiments: erase

We have added to the text to improve the clarity as this is shown extensively throughout the paper. All particles that were eroded from the stoss side and altered the morphology of the dune, were buried before. Those plastic particles were then redeposited and produced comparable outcomes. We have referenced the figures where buried plastic has an impact on the dune morphology, in particular, we encourage you to consider figure 3.

Lines 293-300: I don't think the results of your interesting experiments are of such importance that they allow the recommendation to open a new field of science

We respectfully rebut this point as we think that although there is a lot of work being undertaken by different disciplines in the field of plastic in hydrology, connecting this to sedimentology does result in a new sub-discipline, which will be critical to interdisciplinary communication and advances.

8th Feb 23

Dear Dr Russell,

Please allow me to sincerely apologise for the long delay in sending a decision on your manuscript titled "Plastic pollution in riverbeds fundamentally affects natural sand transport processes". Unfortunately, the previous two reviewers were unable to provide second reports, however, it has now been seen by two further reviewers, whose comments appear below. In light of their advice I am delighted to say that we are happy, in principle, to publish a suitably revised version, in which you provide a clear rationale for your experimental conditions and a more detailed discussion of how representative your experiments are of real river systems, in *Communications Earth & Environment* under the open access CC BY license (Creative Commons Attribution v4.0 International License).

We therefore invite you to revise your paper one last time to address the remaining concerns of our reviewers. At the same time we ask that you edit your manuscript to comply with our format requirements and to maximise the accessibility and therefore the impact of your work.

EDITORIAL REQUESTS:

*****Please take care to match our formatting and policy requirements. We will check revised manuscript and return manuscripts that do not comply. Such requests will lead to delays. *****

SUBMISSION INFORMATION:

OPEN ACCESS:

Communications Earth & Environment is a fully open access journal. Articles are made freely accessible on publication under a [CC BY license](http://creativecommons.org/licenses/by/4.0) (Creative Commons Attribution 4.0 International License). This license allows maximum dissemination and re-use of open access materials and is preferred by many research funding bodies.

For further information about article processing charges, open access funding, and advice and support from Nature Research, please visit <https://www.nature.com/commsenv/article->

processing-charges"><https://www.nature.com/commsenv/article-processing-charges>

At acceptance, you will be provided with instructions for completing this CC BY license on behalf of all authors. This grants us the necessary permissions to publish your paper. Additionally, you will be asked to declare that all required third party permissions have been obtained, and to provide billing information in order to pay the article-processing charge (APC).

[link redacted]

Best regards,

Joe Aslin

Senior Editor,
Communications Earth & Environment
<https://www.nature.com/commsenv/>
Twitter: @CommsEarth

REVIEWERS' COMMENTS:

Reviewer #3 (Remarks to the Author):

Review of Russell et al.

by Tim van Emmerik, Wageningen University, tim.vanemmerik@wur.nl

The paper by Russell et al. presents the hypothesis that that plastic items/particles can change river bedforms and sand transport processes. The work is based on innovative flume experiments that allowed to analyze the transport and accumulation processes in great detail. The paper is generally well-written and thought provoking, and has the potential for publication in Communications Earth & Environment. I have three main points that I hope the authors can address in a revised manuscript.

1. Rationale for the choice of flow conditions and plastic characteristics

The manuscript lacks a clear rationale for the choice of the river flow conditions and the chosen plastic characteristics. What natural rivers does the setup represent? How many of the worlds rivers does it cover? How much of the estimated river plastic emissions does it cover? Similarly, more rationale of the chosen plastic characteristics are necessary. For example, spherical items do not

represent commonly found macroplastics (>0.5 cm) in river systems. If the authors aimed to represent microplastics, I find them relatively large, especially when potentially upscaled to natural conditions.

2. Lack of quantitative results to support the claims

The analysis is highly interesting and innovative. However, I think the manuscript would benefit from more quantitative analysis. The authors describe changes in many relevant variables (e.g. “stoss side becomes significantly over steepened”, “the rate of erosion”, “plastic to sand ratio”, “smaller particles were found to have been distributed more generally”), but I find it difficult to put this into perspective without any numerical values. Also when the authors claim that the plastics “change” sand transport processes, it would be valuable to present some quantitative support for this.

3. Make the implications more quantitative

The implications and effects are often described, but not quantified. The authors suggest that the resistance and river stage may be changed locally. Can the authors provide some estimates of how much the effect would be, based on observed plastic concentrations from literature? The authors hypothesize that plastics interact profoundly with sediment. Can they provide a quantitative comparison between the values of relevant variables with and without interactions with the plastics?

Specific comments:

In the rebuttal the authors refer Paolo et al. (2009) numerously. Is there any other relevant work that can be used as support for the authors' choices?

It is not clear how the 13 different plastic types were selected. Do they represent the most commonly found items in specific rivers? How were the dimensions determined? How do the item characteristics scale to (small urban) rivers at the actual scale?

How were the flow conditions determined, and how do they scale to natural rivers?

The authors detail that the plastic concentration used in the experiments was low, but I cannot find what concentration was used. Also it would be valuable to add what the concentration was based on. Finally, did the concentration remain the same (items added during the experiment), or did the concentration decrease (no items added, while items interacted with the bed)? In case of the latter, what portion of the introduced plastics were deposited/buried? In natural systems, there is a constant supply of plastic from upstream. Do the authors expect different behavior in natural systems compared to the flume experiments?

Fig 1 has great potential and the drawings are very clear. However, I find the text too overwhelming. I suggest that the texts can be reduced to single words, and the description in the caption can be extended.

I agree with reviewer 2 that the statements on the effect on sand migration need more support. Can

the authors estimate how the sand migration/sand transport changed between the case with and without plastic?

I find that the sections “Key findings” and “Challenges and opportunities” contain very few references to relevant work. The authors should provide sufficient literature support for their statements, and link their recommendations to more recent relevant work.

The authors only refer to Morrit et al. (2014) for plastic concentrations in the water column. However, there are plenty of recent papers that have quantified plastics below the surface. See e.g. Liedermann et al. (2018), Lenaker et al. (2019), Haberstroh (2021a, 2021b), Schöneich-Argent et al. (2019), Blondel and Buschman (2022). I recommend to compare the used concentrations in the experiments to the available field observations, to clarify what natural conditions the experiments represent.

Line 30: Plastic and sand interactions  Plastic and riverbed interactions?

Line 38: The 12 MT from Jambeck et al. (2015) is not only rivers, so please rephrase to “land inputs into the ocean” or present one of the global estimates of river plastic emission (e.g. Meijer et al., (2021); Lebreton et al., (2017))

Line 60-69: I suggest to add examples of the spatial and/or temporal scale of the effects of these “small-scale localized changes”.

Line 75: Would be relevant to specify what the authors mean with “far more variable”. Do they mean item/particle heterogeneity, such as discussed in papers by De Lange et al. (2023) and Kooi et al. (2019)?

Line 96: What defines “enough”? Reviewers also commented on this, and I agree it would be valuable to understand how this was determined.

Line 104: What are the dimensions/characteristics of the “small urban streams” that the experiments such represent?

Line 114: To what extent do the authors expect that plastic transport dynamics are also scale independent? Honingh et al. (2020) discussed the scaling challenges of macroplastics in flume experiments, and I am interested to read more about the authors view on this.

Line 122: I think more support is required to make clear why plastics are special or different, compared to other sediments, particles or pollution types. Should the sub-branch focus on plastic-sediment interactions only, or also include other pollution or material types?

Lines 327-328: The authors should clarify what they mean with the “0.12% by mass”, and “0.081% by mass”.

Line 377: Can you expand on the scaling challenges for plastic?

Line 387: Can you provide some specific suggestions on what future work should focus on?

Line 393: How is the work related to environmental monitoring? This link was not clearly made before.

Line 415: How many items/particles were added?

Line 424: Can you provide references that support that you used generic plastic types? For macroplastics, rounded shapes are rarely found for example.

References

Blondel, E., & Buschman, F. A. (2022). Vertical and Horizontal Plastic Litter Distribution in a Bend of a Tidal River. *Frontiers in Environmental Science*, 587.

de Lange SI, Mellink Y, Vriend P, Tasseront PF, Begemann F, Hauk R, Aalderink H, Hamers E, Jansson P, Jooze N, Löhr AJ, Lotcheris R, Schreyers L, Vos V and van Emmerik THM (2023) Sample size requirements for riverbank macrolitter characterization. *Front. Water* 4:1085285. doi: 10.3389/frwa.2022.1085285

Haberstroh, C. J., Arias, M. E., Yin, Z., & Wang, M. C. (2021a). Effects of hydrodynamics on the cross-sectional distribution and transport of plastic in an urban coastal river. *Water Environment Research*, 93(2), 186-200.

Haberstroh, C. J., Arias, M. E., Yin, Z., Sok, T., & Wang, M. C. (2021b). Plastic transport in a complex confluence of the Mekong River in Cambodia. *Environmental Research Letters*, 16(9), 095009.

Honingh, D., Van Emmerik, T., Uijttewaal, W., Kardhana, H., Hoes, O., & Van de Giesen, N. (2020). Urban river water level increase through plastic waste accumulation at a rack structure. *Frontiers in earth science*, 8, 28.

Kooi, M., & Koelmans, A. A. (2019). Simplifying microplastic via continuous probability distributions for size, shape, and density. *Environmental Science & Technology Letters*, 6(9), 551-557.

Lenaker, P. L., Baldwin, A. K., Corsi, S. R., Mason, S. A., Reneau, P. C., & Scott, J. W. (2019). Vertical distribution of microplastics in the water column and surficial sediment from the Milwaukee River Basin to Lake Michigan. *Environmental science & technology*, 53(21), 12227-12237.

Liedermann, M., Gmeiner, P., Pessenlehner, S., Haimann, M., Hohenblum, P., & Habersack, H. (2018). A methodology for measuring microplastic transport in large or medium rivers. *Water*, 10(4), 414.

Schöneich-Argent, R. I., Dau, K., & Freund, H. (2020). Wasting the North Sea?—A field-based assessment of anthropogenic macrolitter loads and emission rates of three German tributaries. *Environmental Pollution*, 263, 114367.

Reviewer #4 (Remarks to the Author):

Review Summary

The reviewed manuscript presents the results of a flume study on mixed sand and plastic particle transport. Although limited in scope, the novel nature of the study and the broad implications of the results make this an exciting and important contribution to the field of fluvial particle transport. The study is primarily reported in terms of process-oriented interpretations of qualitative observations. I think this is appropriate given the fact that this is the first experimental study on mineral sediment-plastic particle interactions in fluvial channel beds, and that the transport interpretations are appropriately described and vetted within the cannon of fluvial sediment transport. Overall the manuscript is very well written and organized, I appreciate the thorough attention to previous reviewer comments which appear to have improved the manuscript. In particular, the authors have appropriately considered the limitations of the study in terms of scale, plastic concentration, and complexity. For these reasons I recommend the article for publication. I have made a few minor suggestions below and look forward to seeing this manuscript in print.

-Andrew Gray
UC Riverside

Detailed Comments

Line/Section Comment

40 Jambeck et al. (2015) is a little dated. See Meijer et al (2021) for a more recent estimate of global fluvial discharge of plastics to the ocean.

42-43 Tasserson et al. (2020) is more of a standing stock evaluation methods paper. Other papers more relevant to fluvial transport include Cowger et al. (2021), Wright et al., (2022), and Valero et al. (2022).

140 D is defined here vaguely as size, but more specifically as equivalent diameter in the methods section, and should include units.

145-146 No need to define R and D again. Also, please include the units of RD, which after reading the Methods section I see are indeed in units of mm. Since the mobility of plastic particles relative to the dominant bed material (sand) is important, why not report instead the unitless 'RD ration to sand,' which was included in the Methods section?

374-379 Agreed, and even in small rivers and streams the effects on flux rates are likely to be less than these preliminary flume experiments if one considers that effective discharge conditions will in most cases be much swifter and deeper flow.

402 It would be nice to have a metric for sorting as well.

References

Cowger W, Gray AB, Guilinger JJ, Fong B, Waldschläger K. 2021. Environmental Science & Technology, 55 (9), 6032-6041. <https://doi.org/10.1021/acs.est.1c01768>

Valero D, Belay BS, Moreno-Rodenas A, Kramer M, Franca MJ. 2022. The key role of surface tension in the transport and quantification of plastic pollution in rivers. *Water Research*, 226: 119078. DOI: <https://doi.org/10.1016/j.watres.2022.119078>.

Wright K, Hariharan J, Passalacqua P, Salter G, Lamb, MP. 2022. From grains to plastics: Modeling nourishment patterns and hydraulic sorting of fluvially transported materials in deltas. *Journal of Geophysical Research: Earth Surface*, 127, e2022JF006769. <https://doi.org/10.1029/2022JF006769>

REVIEWERS' COMMENTS:

Reviewer #3 (Remarks to the Author):

Review of Russell et al.

by Tim van Emmerik, Wageningen University, tim.vanemmerik@wur.nl

The paper by Russell et al. presents the hypothesis that that plastic items/particles can change river bedforms and sand transport processes. The work is based on innovative flume experiments that allowed to analyze the transport and accumulation processes in great detail. The paper is generally well-written and thought provoking, and has the potential for publication in Communications Earth & Environment. I have three main points that I hope the authors can address in a revised manuscript.

1. Rationale for the choice of flow conditions and plastic characteristics

The manuscript lacks a clear rationale for the choice of the river flow conditions and the chosen plastic characteristics. What natural rivers does the setup represent? How many of the worlds rivers does it cover? How much of the estimated river plastic emissions does it cover? Similarly, more rationale of the chosen plastic characteristics are necessary. For example, spherical items do not represent commonly found macroplastics (>0.5 cm) in river systems. If the authors aimed to represent microplastics, I find them relatively large, especially when potentially upscaled to natural conditions.

Lines 104 – 109 already provide an answer to the kinds of rivers this study represents and the types of plastics used. However, the near-bed processes observed are likely to occur in other streams as well and studying 'real-world' processes in laboratory settings has proven to have an unreasonable-effectiveness. Paola et al.(2009) summarize over 30 years of such experiments and explains their potential, even in the absence of real-world analogues for which experiments are specifically designed. A focus on the processes (not on specific systems) enables us to learn far more.

Additionally, there is very little data on the plastic materials found on the bottom of rivers, so we do not have that data as a proxy. Where we do have data, it is European-centric, and highly variable over time, even in the same reach, therefore, it is through focussing on the processes that we can learn more in the most efficient way. Lastly, we did not aim to represent only microplastics as we state on lines 100 and 101, we intentionally included a mixture of plastic materials of different densities and sizes.

Regarding how many of the world's rivers this covers, dunes and ripples occur in all fluvial systems with available substrate, and the scale of these dunes depends entirely on where in the channel you are located, and the present rate of flow of the channel, which is variable. A range of dune scales may be found within a single river bend. So, the data required to determine how many rivers this represents does not exist. The novelty in this study is showing that there is indeed a process that occurs, which is until now, not considered. Therefore, we emphasise that it is the process that requires the study and this is what we have done in this manuscript.

2. Lack of quantitative results to support the claims

The analysis is highly interesting and innovative. However, I think the manuscript would benefit from more quantitative analysis. The authors describe changes in many relevant variables (e.g. “stoss side becomes significantly over steepened”, “the rate of erosion”, “plastic to sand ratio”, “smaller particles were found to have been distributed more generally”), but I find it difficult to put this into perspective without any numerical values. Also when the authors claim that the plastics “change” sand transport processes, it would be valuable to present some quantitative support for this.

We agree with the reviewer that quantifying these processes is extremely important to further develop this research area. Our data measurement techniques for these set of experiments prevent us from taking a more systematic approach at quantifying the observed processes and we wanted to publish these qualitative findings with enough evidence (see Figures and Videos) to promote more research studies and contribute to make the community aware of these things for future studies.

Where we outline rates of change, we do so in a relative and local context, thereby keeping the results and findings within the realm of observation.

3. Make the implications more quantitative

The implications and effects are often described, but not quantified. The authors suggest that the resistance and river stage may be changed locally. Can the authors provide some estimates of how much the effect would be, based on observed plastic concentrations from literature? The authors hypothesize that plastics interact profoundly with sediment. Can they provide a quantitative comparison between the values of relevant variables with and without interactions with the plastics?

We agree with the reviewer that quantifying these processes is extremely important to further develop this research area. Our data measurement techniques for these set of experiments prevent us from taking a more systematic approach at quantifying the observed processes and we wanted to publish these qualitative findings with enough evidence (see Figures and Videos) to promote more research studies and contribute to make the community aware of these things for future studies.

There are no studies that observe plastic in dunes in nature on a river bed, therefore it would be increasing the uncertainty of the findings to begin to draw more specific findings as the comparable data doesn't exist. It is critical that because we are studying processes on the riverbed in sandy dunes, that data on plastic volume comes exclusively from these environments too. We address this in the paragraph beginning line 421.

Lines 331-338 outline how we need more data and broader-scale modelling to answer the questions on what the larger-scale effects would be and how much material would travel downstream as a result.

Specific comments:

In the rebuttal the authors refer Paolo et al. (2009) numerously. Is there any other relevant work that can be used as support for the authors' choices?

Paola et al summarizes over 30 years of lab-scale experiments looking at geomorphic, stratigraphic and sedimentologic experiments. Their effectiveness is more than clear. There is also a limit to the

number of citations and therefore we chose the most representative citations for our manuscript.

It is not clear how the 13 different plastic types were selected. Do they represent the most commonly found items in specific rivers? How were the dimensions determined? How do the item characteristics scale to (small urban) rivers at the actual scale?

The scale of the particles and the urban streams that motivated the study are actually one-to-one and the flow velocities used and flow conditions chosen also match them. However, our findings are likely to extend beyond these small systems as the near-bed particle-to-particle interactions are the most relevant issue impacting overall changes. As outlined above, there is very little data on the plastic materials found on the bottom of rivers, so we do not have that data as a proxy. Where we do have data, it is European-centric. We outline in lines 426-442 why these types of plastic particle have been chosen in the experiment. This is broadly for the understanding of the novel processes that we outline in the manuscript as the data studies of river bottom plastic interacting with river dunes does not exist, so we have no grounding proxies that can be used.

How were the flow conditions determined, and how do they scale to natural rivers?

Focusing on the processes does not require that a lab-based system match vis-à-vis a system in the real-world. We strongly encourage the reviewer to read Paola et al. (2009) to understand the value of non-scaled experiments while being able to observe morphodynamic similarity.

The authors detail that the plastic concentration used in the experiments was low, but I cannot find what concentration was used. Also it would be valuable to add what the concentration was based on. Finally, did the concentration remain the same (items added during the experiment), or did the concentration decrease (no items added, while items interacted with the bed)? In case of the latter, what portion of the introduced plastics were deposited/buried? In natural systems, there is a constant supply of plastic from upstream. Do the authors expect different behavior in natural systems compared to the flume experiments?

The flume is a closed and recirculating system. The concentration added to the flume is available in the methods on line 419 (0.12% by mass). The concentration that remained and interacted with the bedforms was smaller (see figures videos that show material buried). We also outline the behaviour of the plastic in lines 363-367, wherein we say that all of the plastic was added at the beginning of the experiment, yet these mechanics continued throughout, and we consider the temporal continuation of this.

Fig 1 has great potential and the drawings are very clear. However, I find the text too overwhelming. I suggest that the texts can be reduced to single words, and the description in the caption can be extended.

We will let the post-acceptance experts at Comms E&E let us know what they think and adapt the figure accordingly.

I agree with reviewer 2 that the statements on the effect on sand migration need more support. Can the authors estimate how the sand migration/sand transport changed between the case with and without plastic?

This has been resolved by explaining that the effects can be visually seen and the differences and distinction with and without plastic can be seen in the figures and video, as well as explained in the text.

I find that the sections “Key findings” and “Challenges and opportunities” contain very few references to relevant work. The authors should provide sufficient literature support for their statements, and link their recommendations to more recent relevant work.

Key findings is a review of the key findings of the manuscript. There are references where needed but it would be inappropriate to add more because it is the studies conclusion. Challenges and Opportunities are similar in that they are referenced where needed, a reference has been added to justify that understanding plastic is a developing research field.

The authors only refer to Morrit et al. (2014) for plastic concentrations in the water column. However, there are plenty of recent papers that have quantified plastics below the surface. See e.g. Liedermann et al. (2018), Lenaker et al. (2019), Haberstroh (2021a, 2021b), Schöneich-Argent et al. (2019), Blondel and Buschman (2022). I recommend to compare the used concentrations in the experiments to the available field observations, to clarify what natural conditions the experiments represent.

Thank you for these recommendations, however, both Liedermann et al., 2018 and Lenaker et al., 2019 exclusively focus on microplastics, which is of limited use to this project. Blondel and Buschman (2022) use nets under the surface, which leads to similar challenges as we discuss in the manuscript that the Morrit et al., (2014) experiments did. The Haberstroh papers cite that non-spherical particles are much more complex to understand than spherical particles, which again speaks to our decision to keep the majority of particles in this experiment, focussed on the fundamentals of river and sand interactions, spherical. Additionally, Schöneich-Argent et al. (2019) calls for ground-truth model estimates, highlighting that we don't have good ones, and Haberstroh et al., (2021), also state “We found advective fluxes varied between one to two orders of magnitude, suggesting that large errors can be introduced when monthly or even annual loads are estimated only based on single grab samples, especially when multiplied with a rough estimate of monthly river discharge.”. Therefore, the papers that you suggest simply strengthen the necessity for this manuscript herein, and we are unable to add references unless we are replacing them as there is a reference limit for the journal.

Line 30: Plastic and sand interactions  Plastic and riverbed interactions?

Changed to plastic and riverbed sand interactions

Line 38: The 12 MT from Jambeck et al. (2015) is not only rivers, so please rephrase to “land inputs into the ocean” or present one of the global estimates of river plastic emission (e.g. Meijer et al., (2021); Lebreton et al., (2017))

Ammended

Line 60-69: I suggest to add examples of the spatial and/or temporal scale of the effects of these “small-scale localized changes”.

For the nature of this short-form manuscript, it would be inappropriate to go into more detail here. We feel that there are sufficient references included for the curious reader to find out more if required.

Line 75: Would be relevant to specify what the authors mean with “far more variable”. Do they mean item/particle heterogeneity, such as discussed in papers by De Lange et al. (2023) and Kooi et al. (2019)?

We feel that the meaning is clear as it stands

Line 96: What defines “enough”? Reviewers also commented on this, and I agree it would be valuable to understand how this was determined.

The full sentence says that it is enough to create a 10 cm thick deposit along the flume. We have rearranged the sentence to avoid confusion.

Line 104: What are the dimensions/characteristics of the “small urban streams” that the experiments such represent?

This has been covered in discussion above.

Line 114: To what extent do the authors expect that plastic transport dynamics are also scale independent? Honingh et al. (2020) discussed the scaling challenges of macroplastics in flume experiments, and I am interested to read more about the authors view on this.

Scaling is a long-standing challenge in flume experimentation. The paper Paola et al., 2009 summarises over 30 years of discussion on this topic and many others relating to how real world processes can be created in the lab.

Line 122: I think more support is required to make clear why plastics are special or different, compared to other sediments, particles or pollution types. Should the sub-branch focus on plastic-sediment interactions only, or also include other pollution or material types?

The word “plastic” has been replaced with “anthropogenic” to cover this.

Lines 327-328: The authors should clarify what they mean with the “0.12% by mass”, and “0.081% by mass”.

This is clear as phrased and further clarification may be found at lines 418-420 in the Methods.

Line 377: Can you expand on the scaling challenges for plastic?

This is beyond the scope of this paper as here we discuss the novel processes that we have found between sand and plastic. Scaling is not an issue in this case.

Line 387: Can you provide some specific suggestions on what future work should focus on?

To provide specific suggestions would exceed the word count of this manuscript.

Line 393: How is the work related to environmental monitoring? This link was not clearly made

before.

This has been made clearer by adding a few words to the Introduction

Line 415: How many items/particles were added?

All information for this may be found in Table 1, which is now referenced in this part of the manuscript.

Line 424: Can you provide references that support that you used generic plastic types? For macroplastics, rounded shapes are rarely found for example.

The justification for why these plastic particles were selected are outlined clearly in the paragraph lines 423-434 and comment on the data available from the papers you recommend is discussed above.

References

Blondel, E., & Buschman, F. A. (2022). Vertical and Horizontal Plastic Litter Distribution in a Bend of a Tidal River. *Frontiers in Environmental Science*, 587.

de Lange SI, Mellink Y, Vriend P, Tasseron PF, Begemann F, Hauk R, Aalderink H, Hamers E, Jansson P, Joosse N, Löhr AJ, Lotcheris R, Schreyers L, Vos V and van Emmerik THM (2023) Sample size requirements for riverbank macrolitter characterization. *Front. Water* 4:1085285. doi: 10.3389/frwa.2022.1085285

Haberstroh, C. J., Arias, M. E., Yin, Z., & Wang, M. C. (2021a). Effects of hydrodynamics on the cross-sectional distribution and transport of plastic in an urban coastal river. *Water Environment Research*, 93(2), 186-200.

Haberstroh, C. J., Arias, M. E., Yin, Z., Sok, T., & Wang, M. C. (2021b). Plastic transport in a complex confluence of the Mekong River in Cambodia. *Environmental Research Letters*, 16(9), 095009.

Honingh, D., Van Emmerik, T., Uijtewaal, W., Kardhana, H., Hoes, O., & Van de Giesen, N. (2020). Urban river water level increase through plastic waste accumulation at a rack structure. *Frontiers in earth science*, 8, 28.

Kooi, M., & Koelmans, A. A. (2019). Simplifying microplastic via continuous probability distributions for size, shape, and density. *Environmental Science & Technology Letters*, 6(9), 551-557.

Lenaker, P. L., Baldwin, A. K., Corsi, S. R., Mason, S. A., Reneau, P. C., & Scott, J. W. (2019). Vertical distribution of microplastics in the water column and surficial sediment from the Milwaukee River Basin to Lake Michigan. *Environmental science & technology*, 53(21), 12227-12237.

Liedermann, M., Gmeiner, P., Pessenlehner, S., Haimann, M., Hohenblum, P., & Habersack, H. (2018). A methodology for measuring microplastic transport in large or medium rivers. *Water*, 10(4), 414.

Schöneich-Argent, R. I., Dau, K., & Freund, H. (2020). Wasting the North Sea?—A field-based

assessment of anthropogenic macrolitter loads and emission rates of three German tributaries. *Environmental Pollution*, 263, 114367.

Reviewer #4 (Remarks to the Author):

Review Summary

The reviewed manuscript presents the results of a flume study on mixed sand and plastic particle transport. Although limited in scope, the novel nature of the study and the broad implications of the results make this an exciting and important contribution to the field of fluvial particle transport. The study is primarily reported in terms of process-oriented interpretations of qualitative observations. I think this is appropriate given the fact that this is the first experimental study on mineral sediment-plastic particle interactions in fluvial channel beds, and that the transport interpretations are appropriately described and vetted within the cannon of fluvial sediment transport. Overall the manuscript is very well written and organized, I appreciate the thorough attention to previous reviewer comments which appear to have improved the manuscript. In particular, the authors have appropriately considered the limitations of the study in terms of scale, plastic concentration, and complexity. For these reasons I recommend the article for publication. I have made a few minor suggestions below and look forward to seeing this manuscript in print.

-Andrew Gray
UC Riverside

Detailed Comments

Line/Section Comment

40 Jambeck et al. (2015) is a little dated. See Meijer et al (2021) for a more recent estimate of global fluvial discharge of plastics to the ocean.

Ammended

42-43 Tasserson et al. (2020) is more of a standing stock evaluation methods paper. Other papers more relevant to fluvial transport include Cowger et al. (2021), Wright et al., (2022), and Valero et al. (2022).

Ammended, replaced Tasserson et al., 2020 with Cowger et al., 2021

140 D is defined here vaguely as size, but more specifically as equivalent diameter in the methods section, and should include units.

Ammended

145-146 No need to define R and D again. Also, please include the units of RD, which after reading the Methods section I see are indeed in units of mm. Since the mobility of plastic particles relative to the dominant bed material (sand) is important, why not report instead the unitless 'RD ration to sand,' which was included in the Methods section?

We have added the units for RD. We prefer to leave the full explanation in the manuscript to avoid mixing R for water with the R for sand. The number suggested from $R_{\text{plastic}} / R_{\text{sand}}$, would result in a number that would not add value to the narrative of the manuscript.

374-379 Agreed, and even in small rivers and streams the effects on flux rates are likely to be less than these preliminary flume experiments if one considers that effective discharge conditions will in most cases be much swifter and deeper flow.

Noted

402 It would be nice to have a metric for sorting as well.

We are not sure what is meant here. The sorting may be read from Figure 1D.

References

Cowger W, Gray AB, Guilinger JJ, Fong B, Waldschläger K. 2021. Environmental Science & Technology, 55 (9), 6032-6041. <https://doi.org/10.1021/acs.est.1c01768>

Valero D, Belay BS, Moreno-Rodenas A, Kramer M, Franca MJ. 2022. The key role of surface tension in the transport and quantification of plastic pollution in rivers. Water Research, 226: 119078. DOI: <https://doi.org/10.1016/j.watres.2022.119078>.

Wright K, Hariharan J, Passalacqua P, Salter G, Lamb, MP. 2022. From grains to plastics: Modeling nourishment patterns and hydraulic sorting of fluvially transported materials in deltas. Journal of Geophysical Research: Earth Surface, 127, e2022JF006769. <https://doi.org/10.1029/2022JF006769>